# On Transportability for Structural Causal Bandits

## Abstract

Intelligent agents equipped with causal knowledge can optimize their action spaces to avoid unnecessary exploration. The *structural causal bandit* framework provides a graphical characterization for identifying actions that are unable to maximize rewards by leveraging prior knowledge of the underlying causal structure. While such knowledge enables an agent to estimate the expected rewards of certain actions based on others in online interactions, there has been little guidance on how to transfer information inferred from arbitrary combinations of datasets collected under different conditions—observational or experimental—and from heterogeneous environments. In this paper, we investigate the structural causal bandit with *transportability*, where priors from the source environments are fused to enhance learning in the deployment setting. We demonstrate that it is possible to exploit invariances across environments to consistently improve learning. The resulting bandit algorithm achieves a sub-linear regret bound with an explicit dependence on informativeness of prior data, and it may outperform standard bandit approaches that rely solely on online learning.

## 1 Introduction

The multi-armed bandit (MAB) (Robbins, 1952; Lai and Robbins, 1985; Lattimore and Szepesvári, 2020) problem is a pivotal topic in sequential decision-making studies, where an agent aims to maximize cumulative rewards by repeatedly choosing actions based on observed rewards, balancing the exploration-exploitation trade-off. Traditionally, MAB problems assume independence among rewards of different arms, meaning that the reward obtained from one arm provides no information about the others. This assumption limits its applicability to scenarios where dependencies between actions are common such as clinical trials, healthcare, and advertising.

Integrating causal knowledge into a decision-making process enables an agent to model decision problems with abundant dependency structures (Zhang et al., 2020; Kumor et al., 2021; Zhang and Bareinboim, 2022; Ruan et al., 2024), where structural causal models (SCMs) (Pearl, 2000) have been employed to represent causal relationships among actions, rewards, and other relevant factors such as context and states. This approach enables agents to make informed decisions by considering how each action causally influences the reward through causal pathways (Bareinboim et al., 2024).

Existing works (Bareinboim et al., 2015; Lattimore et al., 2016; Forney et al., 2017) have shown that MAB algorithms with causal knowledge can significantly outperform others that do not account for causal dependencies. Subsequent work has explored various specialized settings by introducing additional structural assumptions, such as the availability of both observational and experimental distributions or linear mechanisms (Lu et al., 2020; Bilodeau et al., 2022; Feng and Chen, 2023; Varici et al., 2023). Specifically, Lee and Bareinboim (2018) formalized the *structural causal bandit* without any parametric assumptions. Building on this, Lee and Bareinboim (2019) extended the framework to accommodate scenarios involving non-manipulable variables.

Although the framework significantly reduces the action space, it still requires a substantial amount of exploration, which can be costly in many applications. An alternative approach to alleviating the cost of active experimentation is to leverage previous experimental records from related environments. The expectation is that informative prior data can help narrow down reward distributions and circumvent the *cold-start* problem of agents, allowing them to converge to optimal actions faster, ultimately achieving a higher cumulative reward, even without incurring any regret. However, discrepancies

across environments are often significant, meaning that the data obtained from source environments may not always be informative or lead to improvements in a target environment.

**Example. (Cardiovascular disease treatment)** Consider a scenario where $Y$ represents *cardiovascular disease*, $W$ *blood pressure*, $X_1$ the intake of an *antihypertensive drug*, $X_2$ the use of an *anti-diabetic drug*, and $U$ unobserved factors (e.g., physical activity levels, diet patterns; Ferrannini and Cushman (2012)). Prior data from *Houston* (source) is available for designing a population-level treatment strategy for cardiovascular disease patients in *Boston* (target), aiming to determine appropriate medications. Fig. 1 graphically illustrates this scenario. Such data can be useful but must be handled with care, especially if the population is

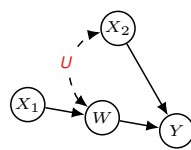

Figure 1: Diagram encoding causal relations.

suspected to differ, e.g., we may expect the distribution $P(U)$ vary across populations. Consider an instance where variables $X_1, X_2, W$, and $Y$ are binary, and their values are determined by functions: $W \leftarrow X_1 \oplus U$, $X_2 \leftarrow U$, and $Y \leftarrow W \wedge X_2$, where $\oplus$ denotes the exclusive-or operation. $X_1$ is drawn independently and uniformly over $\{0, 1\}$ and $P(U{=}1){=}0.4$. Under this system, we find $\mathbb{E}_{P_{do(X_1=1)}} Y = 0.6 > 0.4 = \mathbb{E}_{P_{do(X_1=0)}} Y$ suggesting the antihypertensive drug was effective for Houstonians. Now suppose $P(U{=}1){=}0.7$, reflecting the situation in Boston. In the target, the expected outcome becomes $\mathbb{E}_{P_{do(X_1=1)}} Y = 0.3$, which is the opposite of what was observed in the source. This example illustrates that the optimal strategy in a source can be suboptimal in a target.

In the causality literature, this problem falls under the rubric of *transportability theory* (Pearl and Bareinboim, 2011; Bareinboim and Pearl, 2012b; 2016), which provides methods for determining when and how a causal effect can be computed across different environments. Zhang and Bareinboim (2017) studied the transfer of prior observations in settings with specific graph structures, e.g., Bow and Instrumental Variable (IV), demonstrating that leveraging existing experience can enhance the performance of an agent. Bellot et al. (2023) and Deng et al. (2025) addressed this line of work in general causal diagrams, although their focus was limited to single-node interventions.

**Contributions.** We propose a structural causal bandit algorithm that leverages prior data while accounting for structural discrepancies across heterogeneous environments. Our main contributions are as follows: (1) Within the structural causal bandits framework, we first establish hierarchical relations between action spaces and derive corresponding *dominance* bounds on the expected rewards of actions. (2) We introduce a method for estimating the expected rewards or their *causal* bounds using arbitrary combinations of observational or experimental sources. (3) We provide a UCB-based algorithm that incorporates causal information to guide exploration, and we provide both theoretical guarantees and empirical results showing it achieving a sub-linear cumulative regret depending on the amount of causal knowledge.

## 2 PRELIMINARIES

We introduce notation and review relevant prior work. Following conventions, we use a capital letter, such as $X$, to represent a variable, with its corresponding lowercase letter, $x$, denoting a realization of the variable. Boldface is employed to represent a set of variables or values, denoted by $\mathbf{X}$ or $\mathbf{x}$. The domain of $X$ is indicated by $\Omega_X$ and $\Omega_{\mathbf{X}} = \times_{X \in \mathbf{X}} \Omega_X$. We use calligraphic letters for graphs and models such as $\mathcal{G}$ and $\mathcal{M}$. The distribution over variables $\mathbf{X}$ is denoted by $P(\mathbf{X})$. We consistently use $P(\mathbf{x})$ as an abbreviation for $P(\mathbf{X} = \mathbf{x})$. We denote by $\mathbb{I}\{\mathbf{X} = \mathbf{x}\}$, the indicator function.

**Structural Causal Model.** We use *structural causal model* (SCM) (Pearl, 2000) as the semantic framework to represent the underlying environment a decision maker is deployed. An SCM $\mathcal{M}$ is a quadruple $\langle \mathbf{U}, \mathbf{V}, \mathbf{F}, P(\mathbf{U}) \rangle$, where $\mathbf{U}$ is a set of exogenous variables determined by factors outside the model following a joint distribution $P(\mathbf{U})$, and $\mathbf{V}$ is a set of endogenous variables whose values are determined following a collection of functions $\mathbf{F} = \{f_V\}_{V \in \mathbf{V}}$ such that $V \leftarrow f_V(\mathbf{PA}_V, \mathbf{U}_V)$ where $\mathbf{PA}_V \subseteq \mathbf{V} \setminus \{V\}$ and $\mathbf{U}_V \subseteq \mathbf{U}$. The observational probability $P(\mathbf{v})$ is defined as $\int_{\mathbf{u}} \prod_{V \in \mathbf{V}} \mathbb{I}\{f_V(\mathbf{pa}_V, \mathbf{u}_V) = v\} dP(\mathbf{u})$. Every SCM $\mathcal{M}$ is associated with a *causal diagram* (also called a semi-Markovian graph) $\mathcal{G} = \langle \mathbf{V}, \mathbf{E} \rangle$ where a directed edge $V_i \rightarrow V_j \in \mathbf{E}$ if $V_i \in \mathbf{PA}_{V_j}$, and a bidirected edge between $V_i$ and $V_j$ if $\mathbf{U}_{V_i}$ and $\mathbf{U}_{V_j}$ are correlated. The probability

of $\mathbf{V} = \mathbf{v}$ when $\mathbf{X}$ is intervened upon to take the value $\mathbf{x}$ is denoted by $P(\mathbf{v} \mid do(\mathbf{x}))$ or $P_{\mathbf{x}}(\mathbf{v})$, and the submodel induced by the intervention is denoted by $\mathcal{M}_{\mathbf{x}}$.

**Graphical notations.** An ordered sequence of edges is called a *path*. If a path consists of directed edges with the same orientation, we say the path is *directed*. A path is *directed* from $X$ to $Y$ if there is no arrowhead on the path pointing towards $X$. If there is a (possibly empty) directed path from $X$ to $Y$, then $Y$ is called a *descendant* of $X$, and $X$ is an *ancestor* of $Y$. A variable $Y$ is referred to as a *child* of $X$, and $X$ is a *parent* of $Y$ if they are adjacent and the edge is not directed into $X$. We denote the ancestors, descendants, parents, and children of a given variable as An, De, Pa, and Ch, respectively. Ancestors and descendants include the variable itself. For a set of variables, we define the ancestral set as $\text{An}(\mathbf{X})_{\mathcal{G}} = \bigcup_{X \in \mathbf{X}} \text{An}(X)_{\mathcal{G}}$, and similarly for other relationships. The $\mathbf{X}$-lower-manipulation of $\mathcal{G}$ removes all outgoing edges from variables in $\mathbf{X}$, denoted as $\mathcal{G}_{\underline{\mathbf{X}}}$, while the $\mathbf{X}$-upper-manipulation of $\mathcal{G}$ removes all incoming edges into variables in $\mathbf{X}$ in $\mathcal{G}$, denoted as $\mathcal{G}_{\overline{\mathbf{X}}}$. We denote the set of variables in $\mathcal{G}$ by $\mathbf{V}(\mathcal{G})$. A subgraph $\mathcal{G}[\mathbf{V}']$, where $\mathbf{V}' \subseteq \mathbf{V}(\mathcal{G})$ is defined as a vertex-induced subgraph in which all edges among the vertices in $\mathbf{V}'$ are preserved. We define $\mathcal{G} \setminus \mathbf{X}$ as $\mathcal{G}[\mathbf{V}(\mathcal{G}) \setminus \mathbf{X}]$ for $\mathbf{X} \subseteq \mathbf{V}(\mathcal{G})$. We denote $\mathcal{G}\langle\mathbf{X}\rangle$ as the latent projection of $\mathcal{G}$ on to $\mathbf{X}$. We provide related literature in Appendix A, along with detailed background for our work in Appendix B.

## 3 STRUCTURAL CAUSAL BANDITS WITH TRANSPORTABILITY

We formalize the *structural causal bandit with transportability* problem, where an agent interacts with a target system modeled by a structural causal model (SCM) $\mathcal{M}^*$ including a reward variable $Y \in \mathbf{V}$. In this setting, pulling each arm corresponds to intervening on a set of variables $\{\mathbf{x} \in \Omega_{\mathbf{X}} \mid \mathbf{X} \subseteq \mathbf{V} \setminus \{Y\} \setminus \mathbf{N}\}$ where $\mathbf{N}$ denotes non-manipulable variables. We use the terms *arm*, *action*, and *intervention* interchangeably, depending on the context. The agent cannot access the target system $\mathcal{M}^*$ but can observe $\mathbf{V}$ through online interaction by pulling an arm $do(\mathbf{x})$. In addition, the agent has access to data from one or more related source environments $\Pi = \{\pi^1, \pi^2, \cdots, \pi^n\}$ each associated with SCMs $\mathcal{M}^1, \mathcal{M}^2, \cdots, \mathcal{M}^n$. The distributions associated with $\pi^i$ under $do(\mathbf{x})$ will be denoted by $P_{\mathbf{x}}^i$. We use the superscript $*$ throughout this paper to consistently denote the target environment.

**Graph encoding differences among environments.** To account for environment shift, we introduce a *selection diagram* (Bareinboim and Pearl, 2012b) that captures discrepancies across environments.

**Definition 1** (Environment discrepancy). Let $\pi^i$ and $\pi^j$ be environments associated with SCMs $\mathcal{M}_1$ and $\mathcal{M}_2$ conforming to a causal diagram $\mathcal{G}$. We denote by $\Delta^{i,j}$ a set of variables such that, for every $V \in \Delta^{i,j}$, there exists a discrepancy; either $f_V^i \neq f_V^j$ or $P^i(\mathbf{U}_V) \neq P^j(\mathbf{U}_V)$[1].

**Definition 2** (Selection diagram). Given a collection of discrepancies $\boldsymbol{\Delta} = \{\Delta^{*,i}\}_{i=1}^n$ with regard to $\mathcal{G} = \langle \mathbf{V}, \mathbf{E} \rangle$, let $\mathbf{S}^i = \{S_V \mid V \in \Delta^{*,i}\}$ be *selection nodes*. The graph $\mathcal{G}^{\Delta^{*,i}} = \langle \mathbf{V} \cup \mathbf{S}^i, \mathbf{E} \cup \{S_V \to V\}_{S_V \in \mathbf{S}^i} \rangle$ is called a *selection diagram*. Let $\mathbf{S} = \bigcup_{i=1}^n \mathbf{S}^i$. The *collective selection diagram* $\mathcal{G}^{\boldsymbol{\Delta}}$ is defined as $\langle \mathbf{V} \cup \mathbf{S}, \mathbf{E} \cup \{S_V \to V\}_{S_V \in \mathbf{S}} \rangle$.

We shorten $\Delta^{*,i}$ as $\Delta^i$. The collective selection diagram $\mathcal{G}^{\boldsymbol{\Delta}}$ encodes qualitative information about $\mathcal{M}^*$ and discrepancies $\boldsymbol{\Delta}$. The absence of a selection node pointing to a variable indicates that the causal mechanism responsible for assigning values to that variable is identical across all environments corresponding to $\pi^*$ and $\Pi = \{\pi^i\}_{i=1}^n$. One can view the selection nodes $\mathbf{S}$ as switches controlling environment shifts, and the collective selection diagram $\mathcal{G}^{\boldsymbol{\Delta}}$ as the causal diagram for a unified SCM representing heterogeneous SCMs following $P_{\mathbf{x}}^{\boldsymbol{\Delta}}(\mathbf{y} \mid \mathbf{w}, \mathbf{s}^i = \mathbf{i}, \mathbf{s}^{-i} = \mathbf{0}) = P_{\mathbf{x}}^i(\mathbf{y} \mid \mathbf{w})$ and $P_{\mathbf{x}}^{\boldsymbol{\Delta}}(\mathbf{y} \mid \mathbf{w}, \mathbf{s} = \mathbf{0}) = P_{\mathbf{x}}^*(\mathbf{y} \mid \mathbf{w})$ where $\mathbf{S}^{-i} = \mathbf{S} \setminus \mathbf{S}^i$. This representation enables probabilistic operations across the target environment $\pi^*$ and source environments $\Pi$.

We illustrate in Fig. 2a a collective selection diagram $\mathcal{G}^{\boldsymbol{\Delta}}$ corresponding to the introductory example. In this scenario, the distributions of $W$ and $X_2$ may differ between $\mathcal{M}^*$ and $\mathcal{M}^1$ due to the difference in the distribution of unobserved confounder $U$, which influences both variables. This is consistent with $\boldsymbol{\Delta} = \{\Delta^1 = \{W, X_2\}\}$ and $\mathbf{S} = \mathbf{S}^1 = \{S_W, S_{X_2}\}$. In the diagram, the selection nodes $S_W$ and $S_{X_2}$ are represented as red squares.

---

[1] Superscripts on $\Delta$, such as $\Delta^i$, indicate discrepancies with respect to different environments $\pi^i$, while subscripts like $\Delta_{\mathbf{x}}$ denote suboptimal gaps for the regret analysis for a bandit problem.

We have $P^*(x_1) = P^{\mathbf{\Delta}}(x_1 \mid \mathbf{s} = \mathbf{0}) = P^{\mathbf{\Delta}}(x_1 \mid \mathbf{s} = \mathbf{1}) = P^1(x_1)$ due to the *d-separation* relation (Pearl, 1995) $(\mathbf{S} \perp\!\!\!\perp_d X_1)_{\mathcal{G}^{\mathbf{\Delta}}}$ and a similar equality can be derived for $Y$. In contrast, it may *not* hold $P^*(w, x_2) = P^1(w, x_2)$ due to $(\mathbf{S} \not\perp\!\!\!\perp_d W, X_2)_{\mathcal{G}^{\mathbf{\Delta}}}$. This result indicates that, given access to $P^1(\mathbf{v})$, the probability $P^*(x_1)$ is inferable via $P^1(x_1) = \sum_{\mathbf{v} \setminus \{x_1\}} P^1(\mathbf{v})$, whereas $P^*(w, x_2)$ is not. In this sense, we may say that $P^*(x_1)$ is transportable from the source environment $\pi^1$. We will provide, within the context of structural causal bandits, a formal definition of transportability and study it in detail in Sec. 4.

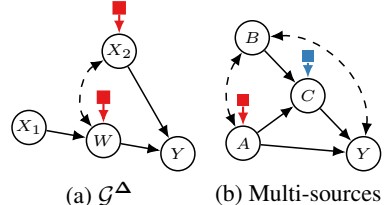

(a) $\mathcal{G}^{\mathbf{\Delta}}$       (b) Multi-sources

Figure 2: Collective selection diagrams for (a) the introductory example and (b) $\Delta^1 = \{A\}$ (red) and $\Delta^2 = \{C\}$ (blue).

**Definition 3** (Structural causal bandits with transportability). Let $\mathbf{x}_t$ be the action taken at round $t \in \{1, \cdots, T\}$. The goal of *structural causal bandits with transportability* is to minimize cumulative regret in the target environment $\pi^*$ defined as follows:

$$R_T = \sum_{t=1}^{T} \mathbb{E}_{P^*_{\mathbf{x}^\star}} Y - \mathbb{E}_{P^*_{\mathbf{x}_t}} Y = \sum_{\mathbf{x}} \Delta_{\mathbf{x}} \mathbb{E} N_T(\mathbf{x}) \tag{1}$$

that compares the reward of the optimal arm $\mathbf{x}^\star = \arg \max_{\mathbf{x} \in \Omega_{\mathbf{X}}, \mathbf{X} \subseteq \mathbf{V} \setminus \{Y\} \setminus \mathbf{N}} \mathbb{E}_{P^*_{\mathbf{x}}} Y$ with that of arm $\mathbf{x}_t$ in each round $t$. $\Delta_{\mathbf{x}}$ denotes suboptimal gap $\mathbb{E}_{P^*_{\mathbf{x}^\star}} Y - \mathbb{E}_{P^*_{\mathbf{x}}} Y$ and $N_T(\mathbf{x})$ denotes the number of times an action $\mathbf{x}$ was chosen up to round $T$.

**Action space worth exploring.** In settings where variables exhibit causal relationships represented by a causal diagram, restricting attention to a subset of the entire action space can lead to improved performance. This implies that instead of exploring all the exponential subsets in $2^{\mathbf{V} \setminus \{Y\} \setminus \mathbf{N}}$, it suffices to consider some subspace. We denote by $\mathbf{x}^* = \arg \max_{\mathbf{x} \in \Omega_{\mathbf{X}}} \mathbb{E}_{P_{\mathbf{x}}} Y$ the best expected reward by intervening on $\mathbf{X}$, and $\mathbf{x}[\mathbf{X}']$ the values of $\mathbf{x}$ restricted to the subset of variables of $\mathbf{X} \cap \mathbf{X}'$.

**Definition 4** (Minimality (Lee and Bareinboim, 2018)). If $\mathbf{X} \subseteq \mathbf{V} \setminus \{Y\} \setminus \mathbf{N}$ be a set such that there is no $\mathbf{X}' \subseteq \mathbf{X}$ such that $\mathbb{E}_{P_{\mathbf{x}}} Y = \mathbb{E}_{P_{\mathbf{x}[\mathbf{X}']}} Y$[2], we refer to $\mathbf{X}$ as a *minimal* intervention set (MIS).

**Definition 5** (Possibly-optimal intervention set). Let $\mathbf{X} \subseteq \mathbf{V} \setminus \{Y\} \setminus \mathbf{N}$ be a set of variables. We say that $\mathbf{X}$ is a *possibly-optimal intervention set* (POIS) with respect to $\langle \mathcal{G}, Y, \mathbf{N} \rangle$ if there exists an SCM conforming to $\mathcal{G}$ such that $\mathbb{E}_{P_{\mathbf{x}^*}} Y > \mathbb{E}_{P_{\mathbf{w}^*}} Y$ for all $\mathbf{W} \subseteq \mathbf{V} \setminus \{Y\} \setminus \mathbf{N}$ that are *not* equivalent to $\mathbf{X}$.

Minimality implies that every variable $X \in \mathbf{X}$ affects the reward variable without passing through $\mathbf{X} \setminus \{X\}$ in $\mathcal{G}$. We refer to a set as a *possibly-optimal minimal intervention set* (POMIS) (Lee and Bareinboim, 2018; 2019) if it is both a POIS and minimal. We denote by $\mathbb{P}^{\mathbf{N}}_{\mathcal{G}, Y}$ a set of POMISs with respect to $\langle \mathcal{G}, Y, \mathbf{N} \rangle$[3]. By definition of POMIS, intervening on non-POMIS cannot yield a better outcome than the optimal one associated with POMIS. This means $\mathbf{x}^\star$ in Eq. (1) can be equivalently expressed as $\mathbf{x}^\star = \arg \max_{\mathbf{x} \in \Omega_{\mathbf{X}}, \mathbf{X} \in \mathbb{P}^{\mathbf{N}^*}_{\mathcal{G}, Y}} \mathbb{E}_{P^*_{\mathbf{x}}} Y$ restricting the exploration space to POMISs. Therefore, an agent who is aware of POMIS should only explore and exploit actions consistent with those sets. All graphical characterizations for PO(M)IS are in Appendix C.

### 3.1 DOMINANCE RELATIONSHIPS AMONG ACTION SPACES

We say an action space *dominates* another when it behaves better than or equal to another with respect to maximum achievable expected reward. For example, it is immediate from the definition of POIS (Def. 5) that all POISs with respect to $\langle \mathcal{G}, Y, \mathbf{N} \rangle$ dominate any non-POISs under the same constraint. Let $\mathbf{W} \subseteq \mathbf{V} \setminus \{Y\} \setminus \mathbf{N}^*$ be a set that is *not* a POIS with respect to $\langle \mathcal{G}, Y, \mathbf{N}^* \rangle$. According to Def. 5, we can derive that $\mathbf{W}$ cannot outperform the target POMISs with respect to $\langle \mathcal{G}, Y, \mathbf{N}^* \rangle$ (i.e., POMISs in the target environment). The following inequality states that *the target POMIS dominates sets which are non-POISs*.

$$\mathbb{E}_{P^*_{\mathbf{x}^\star}} Y \geq \mathbb{E}_{P^*_{\mathbf{w}^*}} Y \tag{2}$$

This inequality implies that if there exists at least one non-POIS action $\mathbf{w} \in \Omega_{\mathbf{W}}$ whose expected reward is greater than that of any target POMIS action, then such an action cannot be the true optimal action. Beyond such trivial cases, we now turn our attention to more nuanced dominance relations

---

[2]We refer to $\mathbf{X}$ and $\mathbf{X}'$ as *equivalent* if the equality holds.

[3]For readability, we omit $\mathbf{N}$ when $\mathbf{N} = \emptyset$ (e.g., $\mathbb{P}_{\mathcal{G}, Y}$ and $\langle \mathcal{G}, Y \rangle$), referring to this case as *unconstrained*.

that arise between constrained and unconstrained PO(M)ISs. Let $\mathbf{r}^\star$ be an optimal POIS action with respect to $\langle \mathcal{G}, Y \rangle$. The following inequality says *unconstrained POISs dominates target POMISs*.

$$\mathbb{E}_{P^*_{\mathbf{x}^\star}} Y \leq \mathbb{E}_{P^*_{\mathbf{r}^\star}} Y. \tag{3}$$

This implies that the expected rewards of target POMIS arms are upper bounded by the right-hand side of Eq. (3). To witness, consider the ongoing cardiovascular example (Fig. 2a). Suppose blood pressure $W$ is non-manipulable, i.e., $\mathbf{N}^* = \{W\}$. The set of POMISs is then given by $\mathbb{P}^{\mathbf{N}^*}_{\mathcal{G},Y} = \{\{X_1\}, \{X_1, X_2\}\}$, which implies that the optimal action $\mathbf{x}^\star$ must be consistent with $do(x_1^*)$ or $do(\{x_1, x_2\}^*)$ but not with $do(\emptyset)$ or $do(x_2^*)$. According to the dominance relationship, $do(x_1)$ and $do(x_1, x_2)$ can be interpreted as the *best alternative* plans under the constraint $\mathbf{N}^* = \{W\}$. For concreteness, we consider an unconstrained (i.e., $\mathbf{N} = \emptyset$) POIS $\mathbf{R} = \{W, X_2\}$. The expected reward under $do(x_1^*)$ can be decomposed as $\mathbb{E}_{P^*_{x_1^*}} Y = \sum_{\mathbf{r}} \mathbb{E}_{P^*_{x_1^*}}[Y \mid \mathbf{r}] P^*_{x_1^*}(\mathbf{r}) \overset{(a)}{=} \sum_{\mathbf{r}} \mathbb{E}_{P^*_{x_1^*, \mathbf{r}}}[Y] P^*_{x_1^*}(\mathbf{r}) \overset{(b)}{=} \sum_{\mathbf{r}} \mathbb{E}_{P^*_{\mathbf{r}}}[Y] P^*_{x_1^*}(\mathbf{r}) \leq \mathbb{E}_{P^*_{\mathbf{r}^*}} Y = \mathbb{E}_{P^*_{\mathbf{r}^\star}} Y$ where (a) follows from Rule 2 and (b) follows from Rule 3 of do-calculus. This inequality shows that $\mathbb{E}_{P^*_{x_1^*}} Y \leq \mathbb{E}_{P^*_{\mathbf{r}^\star}} Y$, and a similar argument applies to $do(\{x_1, x_2\}^*)$ implying that $\mathbb{E}_{P^*_{\mathbf{x}^\star}} Y \leq \mathbb{E}_{P^*_{\mathbf{r}^\star}} Y$. In fact, one can observe $\mathbb{E}_{P^*_{\{x_1, x_2\}^*}} = \mathbb{E}_{P^*_{\mathbf{r}^\star}} Y = \mathbb{E}_{P^*_{x_1^*}} Y = 0.7 \leq \mathbb{E}_{P^*_{\mathbf{r}^\star}} = 1$. We now turn to the dominance relations involving non-POIS actions in Eq. (2). Consider $\mathbf{W} = \{X_2\}$ a non-POIS with respect to $\langle \mathcal{G}, Y, \mathbf{N}^* \rangle$. We find $\mathbb{E}_{P^*_{\mathbf{w}^*}} Y = 0.5$, which implies that the optimal action $\mathbf{x}^\star$ must satisfy $\mathbb{E}_{P^*_{\mathbf{x}^\star}} \geq 0.5$. Indeed, as previously shown, $\mathbb{E}_{P^*_{\mathbf{x}^\star}} = 0.7$. Based on the dominance relations in Eqs. (2) and (3), we introduce the following dominance relationship.

**Theorem 1** (Dominance relationship). *Let $\mathbf{r}^\star$ be an optimal action with respect to $\langle \mathcal{G}, Y, \mathbf{N} \rangle$ where $\mathbf{N}$ is a subset of $\mathbf{N}^*$. Let $\mathbf{W}$ be a non-POIS with respect to $\langle \mathcal{G}, Y, \mathbf{N}^* \rangle$. Then $\mathbb{E}_{P^*_{\mathbf{x}^\star}} Y$ is bounded by*

$$\mathbb{E}_{P^*_{\mathbf{w}^*}} Y \leq \mathbb{E}_{P^*_{\mathbf{x}^\star}} Y \leq \mathbb{E}_{P^*_{\mathbf{r}^\star}} Y. \tag{4}$$

The target POMISs dominate non-POISs under the same constraint, while being dominated by PO(M)ISs defined under a *weaker constraint*. The second inequality in Eq. (4) can be interpreted as a generalized version of Eq. (3) where the target POMISs is regarded as POMISs with respect to $\langle \mathcal{G}\langle \mathbf{V} \setminus \mathbf{N} \rangle, Y, \mathbf{N}^* \setminus \mathbf{N} \rangle$. Therefore, each $\mathbb{E}_{P^*_{\mathbf{r}^\star}} Y$ obtained under a weaker constraint (any subset of $\mathbf{N}^*$) provides a valid upper bound for the target POMISs. While one might be concerned that evaluating POMISs for *all* weaker constraints is computationally exhaustive, it is worth noting that any proper subset of $\mathbf{N}$ corresponds to a strictly weaker constraint than $\mathbf{N}$, and the associated bound is *looser*; and thus the proper

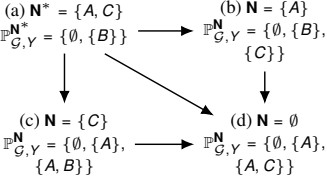

Figure 3: Hierarchical relationships between POMISs under different constraints. Arrows indicate the direction of dominance relations.

subsets of $\mathbf{N}$ dominate $\mathbf{N}$. We illustrate this in Fig. 3, where the target POMISs (a) are defined under $\mathbf{N}^* = \{A, C\}$ for the causal diagram corresponding to Fig. 2b. In this setting, each optimal action under a weaker constraint, (b–d), dominates (a); furthermore, (d) dominates both (b) and (c). This means that (d) is not tighter than either (b) or (c); thus if (b) or (c) is known, estimating (d) is unnecessary to upper bound (a).

Equipping with this dominance knowledge, the corresponding algorithm (Alg. 3 in Appendix D) hierarchically traverses the space of constrained POMISs and stops *early* when a transferable $\mathbb{E}_{P^*_{\mathbf{r}^\star}} Y$ is found, thereby avoiding unnecessary computation. However, it is possible that while $\mathbb{E}_{P^*_{\mathbf{r}^\star}} Y$ is *not* transferable from the sources $\Pi$, its upper bound $u_{\mathbf{r}^\star}$ can still be transferred; thus, it provides valid bounds when integrated with Thm. 1, $\mathbb{E}_{P^*_{\mathbf{x}^\star}} Y \leq u_{\mathbf{r}^\star}$ (we will refer to this as *dominance bounds*). In the following section, we describe how to compute such valid bounds of expected rewards from the sources $\Pi$ (Sec. 4.1) and how to leverage the dominance bounds in online interaction (Sec. 4.2).

## 4 TRANSPORTING ACTIONS FROM SOURCE ENVIRONMENTS

In this section, we investigate transportability of expected rewards in detail and present a method leveraging this knowledge. Consider the collective selection diagram $\mathcal{G}^\Delta$ in Fig. 2b, and suppose that the first dataset from $\pi^1$ is collected under observation $\mathbb{Z}^1 = \{\emptyset\}$ while dataset from $\pi^2$ is obtained under *randomized controlled trial* (RCT) with respect to $\mathbb{Z}^2 = \{\{A, C\}\}$. The target environment $\pi^*$ is assumed to be subject to the constraint $\mathbf{N}^* = \{C\}$. Our objective is to identify transportable

quantities to minimize unnecessary exploration by leveraging prior information from $\pi^1$ and $\pi^2$. As a representative instance, the causal effect of the POMIS action $do(a)$ in $\pi^*$ can be written as:

$$P_a^*(y) = \sum_{b,c} P_a^*(b) P_a^*(c \mid b) P_a^*(y \mid b, c) \overset{(a)}{=} \sum_{b,c} P_{a,c}^*(b) P_\emptyset^*(c \mid a, b) P_{a,c}^*(y \mid b)$$

$$= \sum_{b,c} P_\emptyset^*(c \mid a, b) P_{a,c}^*(b, y) \overset{(b)}{=} \sum_{b,c} P_\emptyset^1(c \mid a, b) P_{a,c}^*(b, y) \overset{(c)}{=} \sum_{b,c} P_\emptyset^1(c \mid a, b) P_{a,c}^2(b, y). \quad (5)$$

Specifically, note that equality (a) follows from applications of Rule 2 (converting conditioning on $c$ to $do(c)$ and $do(a)$ to conditioning on $a$) and Rule 3 (adding $do(c)$) of do-calculus. Equation (b) holds since the discrepancy between $\pi^*$ and $\pi^1$ is irrelevant due to $(S_A \perp\!\!\!\perp_d C \mid A, B)_{\mathcal{G}^\Delta}$. The final equality (c) is derived from the indifference to the disparities in mechanisms $f_B$ and $f_Y$ between $\pi^*$ and $\pi^2$. We observe that the causal effect $P_a^*(y)$ can be expressed in terms of $P_\emptyset^1$ and $P_{a,c}^2$, indicating that $\mathbb{E}_{P_a^*} Y = \sum_y P_a^*(y)$ can be estimated from the sources $\Pi$. Accordingly, we say $P_a^*(y)$ (or equivalently $\mathbb{E}_{P_a^*} Y$) is *transportable*. We now provide a formal definition of transportability.

**Definition 6** (Transportability (Lee et al., 2020)). *Let $\mathcal{G}^\Delta$ be a collective selection diagram with respect to $\Pi = \{\pi_1, \cdots, \pi_n\}$ with a target domain $\pi^*$. Let $\mathbb{Z} = \{\mathbb{Z}^i\}_{i=1}^n$ be a specification of available prior $\mathbb{Z}^i$ conducted in source environment $\pi^i$. We say that $P_{\mathbf{x}}^*(y)$ is transportable with respect to $\langle \mathcal{G}^\Delta, \mathbb{Z} \rangle$ if $P_{\mathbf{x}}^*(y)$ is uniquely computable from $\mathcal{P}_{\mathbb{Z}}^\Pi = \{P_{\mathbf{z}}^i \mid \mathbf{z} \in \Omega_{\mathbf{Z}}, \mathbf{Z} \in \mathbb{Z}^i \in \mathbb{Z}\}$ in any collection of models that induce $\mathcal{G}^\Delta$.*

We introduce the following graphical concepts, which are widely used in the causality literature. Let $cc_\mathcal{G} = \{\mathbf{C}_q\}_{q=1}^m$ be the collection of *c-components* of $\mathcal{G}$. For $\mathbf{C} \subseteq \mathbf{V}$, we define the quantity $Q[\mathbf{C}](\mathbf{v}) = P_{\mathbf{v} \setminus \mathbf{c}}(\mathbf{c})$, which corresponds to the post-intervention distribution and is referred to as a *c-factor*. For convenience, we omit input $\mathbf{v}$ and write $Q[\mathbf{C}]$. We denote the quantities for the target environment $\pi^*$ as $Q^*[\mathbf{C}] = P_{\mathbf{v} \setminus \mathbf{c}}^\Delta(\mathbf{c} \mid \mathbf{s} = \mathbf{0})$ and for sources $\pi^i$ as $Q^i[\mathbf{C}] = P_{\mathbf{v} \setminus \mathbf{c}}^\Delta(\mathbf{c} \mid \mathbf{s}^i = \mathbf{i}, \mathbf{s}^{-i} = \mathbf{0})$ consistently. We denote by $\mathbf{Y}^+ = \text{An}(Y)_{\mathcal{G}_{\mathbf{X}}}$ the set of variables that affect the reward $Y$ under the intervention on $\mathbf{X}$. The expected reward in the target environment $\mathbb{E}_{P_{\mathbf{x}}^*} Y$ can be uniquely expressed using $m$ c-factors $\mathbf{C}_q \in cc_{\mathcal{G}[\mathbf{Y}^+]}$ as follows:

$$\mathbb{E}_{P_{\mathbf{x}}^*} Y = \sum_y y P_{\mathbf{x}}^*(y) = \sum_{\mathbf{y}^+} y \prod_{q=1}^m Q^*[\mathbf{C}_q]. \quad (6)$$

When a c-component $\mathbf{C}$ satisfies $\mathbf{C} \cap \Delta^i = \emptyset$ and there exists a c-component $\mathbf{C} \subseteq \mathbf{C}'$ such that $Q^i[\mathbf{C}]$ is *identifiable*[4] from $\mathcal{G}[\mathbf{C}']$, we refer to $Q^*[\mathbf{C}]$ as being *transportable* from $\pi^i$ (Lee et al., 2020). Furthermore, $P_{\mathbf{x}}^*(Y)$ and $\mathbb{E}_{P_{\mathbf{x}}^*} Y$ are transportable with respect to $\langle \mathcal{G}^\Delta, \mathbb{Z} \rangle$ *if and only if* all c-factors $Q^*[\mathbf{C}_q]$ in the right-hand side of Eq. (6) are transportable from some source environment $\pi^i \in \Pi$. To illustrate, we now reformulate Eq. (5) in terms of c-factors, following Eq. (6): $\mathbb{E}_{P_a^*} Y = \sum_y y P_a^*(y) = \sum_{b,c,y} y Q^*[C] Q^*[B, Y]$. Since $\{C\} \cap \Delta_1 = \emptyset$ and $\{B, Y\} \cap \Delta_2 = \emptyset$, and each corresponding c-factor is identifiable from $\pi^1$ and $\pi^2$, respectively, each c-factor is transportable from $\pi^1$ and $\pi^2$. Therefore, $\mathbb{E}_{P_a^*} Y$ is transportable and is given by $\mathbb{E}_{P_a^*} Y = \sum_{b,c,y} y Q^1[C] Q^2[B, Y]$.

When $\mathbb{E}_{P_{\mathbf{x}}^*} Y$ is not transportable, it may seem that a learning agent cannot obtain any assistance from sources and must estimate outcomes entirely from scratch, engaging in *cold* exploration. However, while determining the exact values may be infeasible, the learner may still extrapolate partial knowledge from the prior to improve estimates within a feasible interval.

### 4.1 BOUNDING NON-TRANSPORTABLE ACTIONS

Our next result concerns the derivation of bounds for the target c-factor $Q^*[\mathbf{C}]$ from a non-identifiable source quantity $Q^i[\mathbf{C}]$ in terms of an identifiable source quantity $Q^i[\mathbf{C}']$ where $\mathbf{C} \subseteq \mathbf{C}'$.

**Proposition 1.** *Let $\mathbf{C}$ be a c-component in $\mathcal{G} \setminus \mathbf{X}$ satisfying $\mathbf{C} \cap \Delta^i = \emptyset$ and $\mathbf{D}$ be a c-component satisfying $\mathbf{C} \subseteq \mathbf{D}$. Let $\mathbf{C}' = \text{An}(\mathbf{C})_{\mathcal{G}[\mathbf{D}]}$. The target c-factor $Q^*[\mathbf{C}]$ is bounded in $[\ell, u]$ where (i) if $\mathbf{C} = \mathbf{C}'$, then $\ell = u = \sum_{\mathbf{c}' \setminus \mathbf{c}} Q^i[\mathbf{C}']$; (ii) otherwise, $\ell = Q^i[\mathbf{C}']$ and $u = Q^i[\mathbf{C}'] + 1 - \sum_{\mathbf{c}} Q^i[\mathbf{C}']$.*

We now revisit the collective selection diagram $\mathcal{G}^\Delta$ in Fig. 2b under a more challenging setting, where the source is modified by supposing that the given distributions correspond to $\mathbb{Z}^1 = \{\emptyset, \{C\}\}$

---

[4] This means that the quantity $Q^i[\mathbf{C}] = P_{\mathbf{v} \setminus \mathbf{c}}^\Delta(\mathbf{c} \mid \mathbf{s}^i = \mathbf{i}, \mathbf{s}^{-i} = \mathbf{0})$ is uniquely computable from $\mathcal{G}[\mathbf{C}']$. We provide the formal definition of identifiability in Def. 7, along with further detailed description in Appendix B.

and $\mathbb{Z}^2 = \{\{B\}\}$. From $\pi^1$, although $\{B, Y\} \cap \Delta_1 = \emptyset$, $Q^1[B, Y]$ is non-identifiable; consequently, $Q^*[B, Y]$ is non-transportable from $\pi^1$. In $\pi^2$, there exists no c-component involving $B$, rendering the target c-factor non-transportable. This structural limitation prevents the learner from identifying $Q^*[B, Y]$, failing transportability of $\mathbb{E}_{P^*}Y$. While $Q^*[B, Y]$ is not transportable from either $\pi^1$ or $\pi^2$, it is important to note that $Q^*[B, Y] = Q^1[B, Y]$ holds due to $\{B, Y\} \cap \Delta_1 = \emptyset$. This implies that any valid bound on $\ell \leq Q^1[B, Y] \leq u$ also induces a valid bound on $Q^*[B, Y]$. Following Prop. 1, we consider $\mathbf{C}' = \{A, B, Y\}$; the lower bound $\ell$ is given by $Q^1[A, B, Y]$, and the upper bound $u$ is $Q^1[A, B, Y] + 1 - \sum_{b,y} Q^1[A, B, Y]$. Therefore, bound on $\mathbb{E}_{P_a^*}Y$ is written as

$$\sum_{b,c,y} y P_\emptyset^1(c \mid a, b) P_c^1(a, b, y) \leq \mathbb{E}_{P_a^*}Y \leq \sum_{b,c,y} y P_\emptyset^1(c \mid a, b) \left\{ P_c^1(a, b, y) + 1 - P_c^1(a) \right\}$$

which is derived by $Q^1[C] = P^1(c \mid a, b)$ and $Q^1[A, B, Y] = P_c^1(a, b, y)$. With this in hand, we are ready to formally construct valid bounds of expected rewards of target actions given $\langle \mathcal{G}^{\Delta}, \mathbb{Z} \rangle$.

**Theorem 2** (Causal bounds). *Given $\langle \mathcal{G}^{\Delta}, \mathbb{Z} \rangle$, the target expected reward $\mathbb{E}_{P_{\mathbf{x}}^*}Y$ can be bounded by $[\ell_{\mathbf{x}}, u_{\mathbf{x}}]$ if for all c-factors $Q^*[\mathbf{C}_q]$ in the right-hand side of Eq. (6), there exists a source $\pi^i \in \Pi$ satisfying $\mathbf{C}_q \cap \Delta^i = \emptyset$ and a computable ancestral c-component $\mathbf{C}_q \subseteq \mathbf{C}_q'$ from $\mathcal{G} \setminus \mathbf{Z}$ where $\mathbf{Z} \in \mathbb{Z}^i \in \mathbb{Z}$. The bound $[\ell_{\mathbf{x}}, u_{\mathbf{x}}]$ is defined as:*

$$\ell_{\mathbf{x}} \triangleq \sum_{\mathbf{y}+} y \prod_{q=1}^{k} Q^{i_q}[\mathbf{C}_q] \prod_{q=k+1}^{m} Q^{j_q}[\mathbf{C}_q'] \leq \sum_{\mathbf{y}+} y \prod_{q=1}^{k} Q^{i_q}[\mathbf{C}_q] \prod_{q=k+1}^{m} Q^*[\mathbf{C}_q] \tag{7}$$

$$\leq \sum_{y} y \min\left\{ 1, \sum_{\mathbf{y}+\setminus\{y\}} \prod_{q=1}^{k} Q^{i_q}[\mathbf{C}_q] \prod_{q=k+1}^{m} \left\{ Q^{j_q}[\mathbf{C}_q'] + 1 - \sum_{\mathbf{c}_q} Q^{j_q}[\mathbf{C}_q'] \right\} \right\} \triangleq u_{\mathbf{x}} \tag{8}$$

*where $\prod_{q=1}^{k} Q^{i_q}[\mathbf{C}_q]$ denotes transportable c-factors.*

The upper bounds of $P_{\mathbf{x}}^*(y)$ may exceed one due to sum-product operations over non-transportable terms. Thus, we take the minimum of the value and one. The corresponding algorithms (Algs. 4 and 5) are presented in Appendix E. First, the algorithm PATR (Alg. 4) outputs *all* expressions for bounds of $P_{\mathbf{x}}^*(y)$, based on Prop. 1 and Thm. 2. Then, CAUSALBOUND (Alg. 5) computes the bounds on $\mathbb{E}_{P_{\mathbf{x}}^*}Y$ over the sources $\Pi$, and returns the tightest lower and upper bounds as the causal bound.

### 4.2 Upper Confidence Bound Algorithm with Transport Bounds

In this section, we now incorporate obtained bounds into a bandit problem. Dominance relationships (Thm. 1) and causal bounds (Thm. 2) can refine the upper confidence bound (UCB) (Auer et al., 2002) estimates during online learning. To illustrate this, we follow the *clipped upper confidence bound* approach (Zhang and Bareinboim, 2017) as our index strategy.[5] We denote the empirical reward estimate at round $t$ for each arm in the target environment $\pi^*$ as $\hat{\mathbb{E}}_{P_{\mathbf{x}}^*, t}Y = \frac{1}{N_t(\mathbf{x})} \sum_{t'=1}^{t} Y_{\mathbf{x}, t'} \mathbb{I}\{\mathbf{X}_{t'} = \mathbf{x}\}$. The standard UCB index is defined as $U_{\mathbf{x}}(t) = \hat{\mathbb{E}}_{P_{\mathbf{x}}^*, t}Y + \sqrt{\frac{\ln(1/\delta)}{2N_t(\mathbf{x})}}$ where $\delta = t^{-4}$. Our index policy is defined as $\bar{U}_{\mathbf{x}}(t) = \min\{\max\{U_{\mathbf{x}}(t), \ell_{\mathbf{x}}\}, u_{\mathbf{x}}\}$ which constrains the standard UCB index within the final transport bounds $[\ell_{\mathbf{x}}, u_{\mathbf{x}}]$. Our algorithm TRUCB is presented in Alg. 1, which begins by initializing the target action space with POMISs: $\mathcal{I}^* \triangleq \{\mathbf{x} \in \Omega_{\mathbf{X}} \mid \mathbf{X} \in \mathbb{P}_{\mathcal{G}, Y}^{\mathbf{N}^*}\}$ (Line 1).[6]

**Dominance bounds and causal bounds.** The next part (Lines 2–4), the algorithm attempts to determine whether the causal bounds can be identified and compute them using Thm. 2 (corresponding to CAUSALBOUND). All causal bounds are initialized as $[0, \infty)$. Using the resulting causal bounds for each action and dominance relationship (Thm. 1), the algorithm computes the *upper dominance bound* $u^*$ by minimizing expected rewards of POMIS actions (or upper causal bounds if non-transportable) for weaker constraints, which may result in a tighter bound. The next step is to obtain the *lower dominance bound* $\ell^*$ by taking the maximum of causal lower bounds $\ell_{\mathbf{w}}$ over minimal actions under the same constraint (i.e., MISs with respect to $\langle \mathcal{G}, Y, \mathbf{N}^* \rangle$). Since such actions can never outperform the optimal arm $\mathbf{x}^*$, we can safely exclude any actions $\mathbf{x} \in \mathcal{I}^*$ whose upper causal bound

---

[5] Our contribution is to characterize which bounds can be derived from causal knowledge in the structural causal bandits, rather than to demonstrate how they can be efficiently exploited in the online phase.

[6] Each subroutine, MISs and POMISs, is an algorithm that returns MISs and POMISs, respectively (Lee and Bareinboim, 2018). Given $\mathcal{G}\langle \mathbf{V} \setminus \mathbf{N}^* \rangle$ and $Y$, these algorithms compute the corresponding sets for $\langle \mathcal{G}, Y, \mathbf{N}^* \rangle$.

---

**Algorithm 1:** TRansport bounds Upper Confidence Bound (TRUCB)

---

**Input:** $Y$: reward; $\mathcal{G}$: causal diagram; $\mathbf{N}^*$: non-manipulable variables; $\boldsymbol{\Delta}$: discrepancies; $\mathbb{Z}$: a specification of priors; $\Pi$: sources; $\mathcal{P}_\mathbb{Z}^\Pi$: available distributions;

1 Initialize the target action space $\mathcal{I}^* \leftarrow \{\mathbf{x} \in \Omega_\mathbf{X} \mid \mathbf{X} \in \mathsf{POMISs}(\mathcal{G}\langle\mathbf{V} \setminus \mathbf{N}^*\rangle, Y)\}$

2 Set the causal bounds $[\ell_\mathbf{x}, u_\mathbf{x}]$ for all actions using CAUSALBOUND (Alg. 5 in Appendix E)

3 Set the upper dominance bound $u^\star$ using UDB (Alg. 3 in App. D); and $u_\mathbf{x} \leftarrow \min\{u_\mathbf{x}, u^\star\}$ for all $\mathbf{x} \in \mathcal{I}^*$.

4 Compute $\ell^\star = \max_{\mathbf{w} \in \Omega_\mathbf{W}, \mathbf{W} \in \mathsf{MISs}(\mathcal{G}\langle\mathbf{V} \setminus \mathbf{N}^*\rangle, Y)} \ell_\mathbf{w}$; and remove actions from $\mathcal{I}^*$ such that $u_\mathbf{x} < \ell^\star$.

5 **for** each trial $t \leq T$ **do**

6     Choose an arm $\mathbf{x}_t = \mathrm{argmax}_{\mathbf{x} \in \mathcal{I}^*} \bar{U}_\mathbf{x}(t)$ where $\bar{U}_\mathbf{x}(t) = \min\{\max\{U_\mathbf{x}(t), \ell_\mathbf{x}\}, u_\mathbf{x}\}$.

7     Intervene on $\mathbf{X}_t = \mathbf{x}_t$ for round $t$ and receive reward $Y_t$ from $P_{\mathbf{x}_t}^*$.

---

$u_\mathbf{x}$ is lower than the lower dominance bound $\ell^\star$. We defer the technical details of the subroutines—CAUSALBOUND and UDB—to Appendices D and E, respectively. After completion of this phase, we refer to $[\ell_\mathbf{x}, u_\mathbf{x}]$ as *transport bounds*, as they incorporate both dominance and causal bounds.

**Clipped UCB.** In the last part (Lines 5–7), the algorithm enters the online interaction phase with the final actions space $\mathcal{I}^*$. At each round $t$, it computes the clipped UCB $\bar{U}_\mathbf{x}(t)$ for every arm by combining the empirical rewards collected up to round $t$ with the transport bounds $[\ell_\mathbf{x}, u_\mathbf{x}]$. The agent then selects an arm with the highest index and receives the corresponding reward. Not surprisingly, this strategy ensures that the cumulative regret grows sublinearly with the number of rounds $T$.

**Proposition 2** (Regret bound). *Let $Y$ be the reward variable supported on $[0,1]$. Then, the cumulative regret of TRUCB in the target SCM $\mathcal{M}^*$ after $T > 1$ rounds is bounded as*

$$R_T \leq 8 \sum_{\mathbf{x}:\Delta_\mathbf{x}>0,\, u_\mathbf{x} \geq \mathbb{E}_{P_{\mathbf{x}^\star}^*} Y} \frac{\log(T)}{\Delta_\mathbf{x}} + \left(1 + \frac{\pi^2}{3}\right) \sum_{\mathbf{x}:\Delta_\mathbf{x}>0,\, u_\mathbf{x} \geq \ell^\star} \Delta_\mathbf{x}. \tag{9}$$

**Corollary 1.** *The expected number of pulls, $\mathbb{E}N_t(\mathbf{x})$, is zero for all actions $\mathbf{x}$ satisfying $u_\mathbf{x} < \ell^\star$.*

*Proof.* All actions $\mathbf{x}$ such that $u_\mathbf{x} < \ell^\star$ are removed from $\mathcal{I}^*$ in Line 4. Let $\bar{\mathbf{x}} = \arg\max_{\mathbf{x} \in \mathcal{I}^*} \ell_\mathbf{x}$. If $u_\mathbf{x} < \ell_{\bar{\mathbf{x}}}$, then for all rounds $t$, it holds that $\bar{U}_\mathbf{x}(t) \leq u_\mathbf{x} < \ell_{\bar{\mathbf{x}}} \leq \bar{U}_{\bar{\mathbf{x}}}(t)$. Hence, any action $\mathbf{x}$ satisfying $u_\mathbf{x} < \ell^\star$ will never be selected throughout the learning process. $\square$

## 5 EXPERIMENTS

In this section, we present empirical results demonstrating that exploration over the action space $\mathcal{I}^*$ and the clipped index within the transport bounds $U_\mathbf{x}(t)$ lead to lower cumulative regret (CR). We compare TRUCB (Alg. 1) with standard UCB over all combinations of arms (UCB) and over POMISs (POUCB), focusing primarily on POUCB to ensure a fair evaluation of transportability. The number of trials is set to 50k, which is sufficient to observe the performance differences, as shown in Fig. 7. Further detailed explanations and settings regarding experiments are provided in Appendix G.

**Task 1.** We start with a simple structural causal bandit problem where $\mathbf{N}^* = \emptyset$ represented by Fig. 4. A decision maker has an RCT prior $\mathbb{Z}^1 = \{\{B\}\}$ from $\Delta^1 = \{A\}$. The action space without prior information corresponds to $\mathbb{P}_{\mathcal{G},Y}^{\mathbf{N}^*} = \{\{B\}, \{A, B\}\}$. In this setting, we have $\mathbb{E}_{P_b^*} Y = \sum_{a,y} yQ^*[A,Y]$ by Eq. (6). However, there is no c-factor satisfying Thm. 2 since $\{A\} \cap \Delta^1 \neq \emptyset$. In contrast, consider $\mathbb{E}_{P_{a,b}^*} Y = \sum_y yQ^*[Y] = \sum_y yQ^1[Y]$ according to

Figure 4

$\{Y\} \cap \Delta^1 = \emptyset$ and we can find $Q^1[A,Y] = P_b^1(a,y)$. This implies that $\sum_y yP_b^1(a,y) \leq \mathbb{E}_{P_{a,b}^*} Y \leq \sum_y y\{P_b^1(a,y)+1-P_b^1(a)\}$. Using these expressions, the decision maker can estimate causal bounds for four actions corresponding to the POMISs: $do(A=0, B=0) : [0.1675, 1]$, $do(A=0, B=1) : [0.2965, 0.7940]$, $do(A=1, B=0) : [0.8325, 1]$ and $do(A=1, B=1) : [0.2935, 0.7960]$. The algorithm computes the dominance bounds as $\ell^\star = 0.8325$ and $u^\star = \infty$. Among the four actions, the upper causal bounds $u_{do(A=0,B=1)}$ and $u_{do(A=1,B=1)}$ are lower than $\ell^\star$, leading them to be pruned by TRUCB. Therefore, the algorithm begins the online interaction with the final action space $\mathcal{I}^*$ excluding these two actions. We observe that the mean cumulative regret at the final trial is **47.94** for TRUCB and **130.16** for POUCB, which is $\frac{\text{CR for TRUCB}}{\text{CR for POUCB}} = \mathbf{36.83\%}$ of the latter.

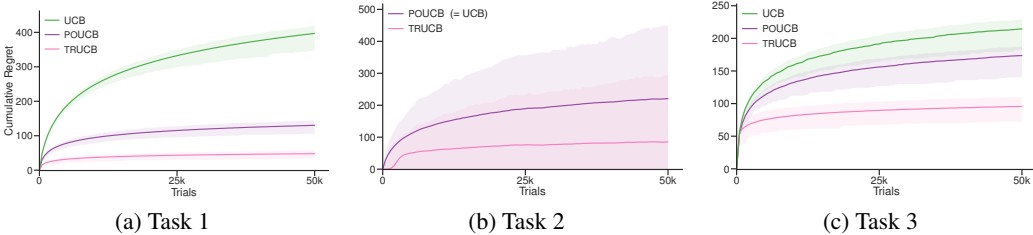

| (a) Task 1 | (b) Task 2 | (c) Task 3 |

Figure 7: Cumulative regrets of TRUCB (pink) compared with standard UCB over all combinations of arms (green) and over POMISs (purple). Each simulation is repeated 1,000 times to ensure consistency, and the shaded regions indicate the 2.5th and 97.5th percentiles of the empirical cumulative regrets.

**Task 2.** We consider the setting in Fig. 5 where $\Delta^1 = \{A\}$ and $\Delta^2 = \{B\}$ with priors $\mathbb{Z}^1 = \{\emptyset\}$ and $\mathbb{Z}^2 = \{\{C\}\}$ and constraint $\mathbf{N}^* = \{A, C\}$. The initial action space corresponds to $\mathbb{P}^{\mathbf{N}^*}_{\mathcal{G},Y}$. The action $do(\emptyset)$ is transportable since $\mathbb{E}_{P^*_\emptyset} Y = \sum_{a,b,c,y} y Q^2[A] Q^1[B, C, Y] = \sum_{a,b,c,y} y P_c^2(a) P^1(b, c, y \mid a) = 0.4844$. On the other hand, $do(b)$ is *not* transportable, as $P_b^*(y) = \sum_{a,c} Q^2[A] Q^2[Y] Q^*[C]$. By Thm. 2, $Q^*[C]$ is bounded by $Q^1[B, C]$, leading to $\ell_b = \sum_{a,c,y} y P_c^2(a) P_c^2(y|a) P^1(c|a, b) P^1(b)$. This expression yields $\ell_{do(B=0)} = 0.2097$ and $\ell_{do(B=1)} = 0.2752$. The upper causal bound for $do(b)$ is given by $\sum_y y \min\{1, \sum_{a,c} P_c^2(a) P_c^2(y|a)\{P^1(b)P^1(c|a, b) + 1 - P^1(b)\}\}$. We thus have the upper causal bounds, $u_{do(B=0)} = 0.6783$ and $u_{do(B=1)} = 0.8066$. The dominance bound is $[\ell^\star, u^\star] = [0.4844, 0.7697]$ with $u^\star = 0.7697$ derived from the expected reward of the transportable action $\mathbb{E}_{P^*_{do(A=1,C=1)}} Y = \sum_y y P^2_{do(C=1)}(y|A=1) = 0.7697$, which is an unconstrained POMIS. Since $u_{do(B=1)}=0.8066 > 0.7697=u^\star$, the final transport upper bound of $do(B=1)$ is updated to $u^\star$. The resulting transport bounds for $do(B=0)$ and $do(B=1)$ are $[0.2097, 0.6783]$ and $[0.2752, 0.7697]$, respectively. Although the size of the action space remains unchanged (i.e., no action is removed from $\mathcal{I}^*$, implying that the action spaces of all three algorithms are identical), we observe that accounting for the transport bounds improves performance, with CR of TRUCB reduced to **38.6**% of POUCB.

Figure 5

**Task 3.** We consider a more involved scenario (Fig. 6) to validate our result. Let $\Delta^1 = \{T\}$ and $\Delta^2 = \{R\}$, with priors $\mathbb{Z}^1 = \{\emptyset, \{Z\}\}$ and $\mathbb{Z}^2 = \{\{Z\}\}$, and constraint $\mathbf{N}^* = \{T, W\}$. The algorithm starts by initializing the action space as $\mathbb{P}^{\mathbf{N}^*}_{\mathcal{G},Y} = \{\emptyset, \{R\}, \{X\}, \{Z\}\}$ where $do(\emptyset)$ and $do(z)$ are transportable while $do(x)$ and $do(r)$ are *not*. In this setting, the transportable target POMIS action yields $\mathbb{E}_{P^*_{do(Z=1)}} Y = 1$, leading to $\ell^\star = u^\star = 1$. Furthermore, we obtain the upper causal bounds $u_{do(\emptyset)} = 0.5514$, $u_{do(X=0)} = 0.7901$ and $u_{do(Z=0)} = 0.034$, which are lower than $\ell^\star$. Consequently, the three actions are eliminated from $\mathcal{I}^*$ by the algorithm. We observe mean cumulative regrets of **173.62** (POUCB)

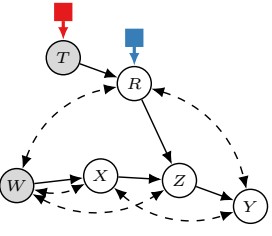

Figure 6

and **95.69** (TRUCB), achieving a **55.1**% regret ratio. These results demonstrate performance improvements when transported causal knowledge from source environments is taken into account.

## 6 CONCLUSION

We investigated a structured causal bandit strategy that can utilize prior data from related heterogeneous environments. Since source environments may differ, some knowledge of the underlying structure and potential discrepancies was necessary to enable consistent extrapolation. To address this, we proposed a strategy that exploits transportable causal knowledge by incorporating bounds equipped with dominance relations and causal structure. We demonstrated that the resulting bandit algorithm, leveraging causal knowledge, enjoys a sub-linear regret bound that depends on the extent of such knowledge. We believe that these results have practical implications for designing intelligent agents, providing a foundation for optimizing the action space when historical data is available.

REPRODUCIBILITY STATEMENT

We respect and fully comply with the ICLR code submission policy. After the discussion forums open, we will post a comment with a link to an anonymous repository. This repository will contain the full code, scripts, and instructions necessary to reproduce all results reported in the paper.

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

CONTENTS

# Appendix for "On Transportability for Structural Causal Bandits"

## A    RELATED WORKS

In this section, we review related work concerned with identification, partial identification, transportability and structural causal bandits.

**Identification.**    Causal effect identification, also known as the *identification problem* (Pearl, 1995), concerns whether the causal effect of an intervention on a set of variables can be uniquely computed from the observational distribution over observed variables and a given causal diagram. Foundational works (Tian and Pearl, 2002; Shpitser and Pearl, 2006; Huang and Valtorta, 2006; 2008) culminated in a complete graphical and algorithmic characterization of the problem. Beyond purely observational sources, there has been growing interest in generalizing the identification problem to settings where both observational and experimental data are available. Bareinboim and Pearl (2012a) studied the conditions under which the causal effect is uniquely computable, given a causal diagram and a collection of observational and experimental distributions over *all* subsets of a given set. Lee et al. (2019) and Kivva et al. (2022) investigated the identification problem over arbitrary collections of distributions and established necessary and sufficient graphical conditions for generalized identification.

**Partial identification.**    Given a causal diagram, one can express a causal effect in terms of the observational distribution using standard identification algorithms. However, challenges related to non-identifiability may arise, and the target effect may not be uniquely computable from observational data. The framework of *partial identification* (Balke and Pearl, 1995; 1997) addresses this issue by constraining the parameter space of causal effects within an interval. Zhang et al. (2022) proposed a polynomial programming approach to solve partial identification problems for arbitrary causal diagrams. Partial identification has also been applied in the causal decision-making literature to estimate dynamic treatment regimes (Zhang and Bareinboim, 2019; 2020; 2022), reinforcement learning (Zhang and Bareinboim, 2024; Bareinboim et al., 2024), bandit algorithms (Zhang and Bareinboim, 2017; 2021; Joshi et al., 2024), and other domains (Jalaldoust et al., 2024; Bellot and Chiappa, 2024; Ruan et al., 2024).

**Transportability.**    In the causal inference literature, the problem of identifying causal effects under potential environment discrepancies has been extensively studied through the theory of *transportability* (Pearl and Bareinboim, 2011; Bareinboim and Pearl, 2012b; Bareinboim et al., 2013; Bareinboim and Pearl, 2016). These works focus on determining whether a causal effect can be identified across environments, and which aspects of causal knowledge can be transferred. Lee et al. (2020) investigated transportability under arbitrary combinations of experiments conducted in both the source and target environments. Correa and Bareinboim (2020) extended this framework to handle soft interventions, while Correa et al. (2022) further generalized it to the setting of counterfactual effects. More recently, Jalaldoust et al. (2024) proposed a parameterization model approach (Zhang et al., 2022; Xia et al., 2021) to bound non-transportable causal effects. Zhang and Bareinboim (2017) explored the integration of prior knowledge into the multi-armed bandit (MAB) framework under restrictive graph structures such as the Bow and IV settings. Building upon this, Bellot et al. (2023) extended these ideas to arbitrary causal diagrams, and Deng et al. (2025) further generalized them to settings where contextual information is available.

**Structural causal bandits.**    Lee and Bareinboim (2018) formalized the *structural causal bandit* framework, in which a bandit instance is structured by an SCM, and each action corresponds to an intervention on a subset of variables. The authors proposed a sound and complete graphical characterization for identifying actions that could be part of an optimal strategy, enabling an agent to avoid unnecessary exploration *a priori*, without any actual interaction. Lee and Bareinboim (2019) extended the framework to settings involving non-manipulable variables. Lee and Bareinboim (2020) established the framework under stochastic policies and demonstrated the informativeness of such policies. Everitt et al. (2021) and Carey et al. (2024) further investigated the completeness of the

graphical characterization of optimal policy spaces, although the general completeness remains an open problem. Wei et al. (2023) proposed a parameterization-based approach to incorporate shared information among possibly-optimal actions. Recently, Elahi et al. (2024) extended the SCM-MAB framework to settings where no causal graph is assumed to be accessible, requiring their algorithm to perform causal discovery—i.e., to construct the causal structure—during online interaction.

# B  BACKGROUND

**D-separation.**  In a causal diagram $\mathcal{G}$, a path $p$ between vertices $X$ and $Y$ is a *d-connecting* path relative to a set $\mathbf{Z}$ if (i) every non-collider on $p$ is not a member of $\mathbf{Z}$; and (ii) every collider on $p$ is an ancestor of some member of $\mathbf{Z}$. Two variables $\mathbf{X}$ and $\mathbf{Y}$ are said to be *d-separated* by $\mathbf{Z}$ if there is no d-connecting path between $X$ and $Y$ relative to $\mathbf{Z}$. Two disjoint sets $\mathbf{X}$ and $\mathbf{Y}$ are said to be d-separated by $\mathbf{Z}$ if every variable in $\mathbf{X}$ is d-separated from every variable in $\mathbf{Y}$ by $\mathbf{Z}$ and denoted as $(\mathbf{X} \perp\!\!\!\perp_d \mathbf{Y} \mid \mathbf{Z})_{\mathcal{G}}$.

**Do-calculus.**  Pearl (1995) devised *do-calculus* which acts as a bridge between observational and interventional distributions from a causal diagram without relying on any parametric assumptions.

**Theorem 3** (Do-calculus (Pearl, 1995))**.** *Let $\mathcal{G}$ be a causal diagram compatible with a structural causal model $\mathcal{M}$, with endogenous variables $\mathbf{V}$. For any disjoint $\mathbf{X}$, $\mathbf{Y}$, $\mathbf{W}$, $\mathbf{Z} \subseteq \mathbf{V}$, the following rules are valid.*

*Rule 1.* $P(\mathbf{y} \mid do(\mathbf{w}), \mathbf{x}, \mathbf{z}) = P(\mathbf{y} \mid do(\mathbf{w}), \mathbf{z})$      *if $\mathbf{X}$ and $\mathbf{Y}$ are d-separated by $\mathbf{W} \cup \mathbf{Z}$ in $\mathcal{G}_{\overline{\mathbf{W}}}$*

*Rule 2.* $P(\mathbf{y} \mid do(\mathbf{w}), do(\mathbf{x}), \mathbf{z}) = P(\mathbf{y} \mid do(\mathbf{w}), \mathbf{x}, \mathbf{z})$      *if $\mathbf{X}$ and $\mathbf{Y}$ are d-separated by $\mathbf{W} \cup \mathbf{Z}$ in $\mathcal{G}_{\overline{\mathbf{W}}, \underline{\mathbf{X}}}$*

*Rule 3.* $P(\mathbf{y} \mid do(\mathbf{w}), do(\mathbf{x}), \mathbf{z}) = P(\mathbf{y} \mid do(\mathbf{w}), \mathbf{z})$      *if $\mathbf{X}$ and $\mathbf{Y}$ are d-separated by $\mathbf{W} \cup \mathbf{Z}$ in $\mathcal{G}_{\overline{\mathbf{W}, \mathbf{X}(\mathbf{Z})}}$*

*where $\mathbf{X}(\mathbf{Z}) \triangleq \mathbf{X} \setminus \mathrm{An}(\mathbf{Z})_{\mathcal{G}[\mathbf{V} \setminus \mathbf{W}]}$.*

**Latent projection.**  The *latent projection* (Verma and Pearl, 1990) of a causal diagram $\mathcal{G}$ over $\mathbf{V}$ on $\mathbf{X}$, denoted by $\mathcal{G}\langle \mathbf{X} \rangle$ is a causal diagram over $\mathbf{X}$ such that, in addition to including edges in $\mathcal{G}[\mathbf{X}]$, for every pair of distinct vertices $V_i, V_j \in \mathbf{X}$, (i) add a directed edge $V_i \rightarrow V_j$ in $\mathcal{G}\langle \mathbf{X} \rangle$ if there exists a directed path from $V_i$ to $V_j$ in $\mathcal{G}$ such that every non-endpoint vertex on the path is not in $\mathbf{X}$, and (ii) add a bidirected edge $V_i \leftrightarrow V_j$ in $\mathcal{G}\langle \mathbf{X} \rangle$ if there exists a divergent path between $V_i$ and $V_j$ in $\mathcal{G}$ such that every non-endpoint vertex on the path is not in $\mathbf{X}$.

**Identification.**  Causal effect identification (Pearl, 1995) concerns whether the causal effect of an intervention on a set of variables can be uniquely computed from the observational distribution over observed variables and a given causal diagram.

**Definition 7** (Identifiability (Pearl, 2000))**.** The causal effect of the intervention on $\mathbf{X} = \mathbf{x}$ is *identifiable* in $\mathcal{G}$, if for any two positive models $\mathcal{M}_1$ and $\mathcal{M}_2$ that induce the causal diagram $\mathcal{G}$, $P^{\mathcal{M}_1}(\mathbf{V}) = P^{\mathcal{M}_2}(\mathbf{V}) > 0$ implies $P_{\mathbf{x}}^{\mathcal{M}_1}(\mathbf{y}) = P_{\mathbf{x}}^{\mathcal{M}_2}(\mathbf{y})$.

In the identification problem, a basic structural unit known as the *c-component (confounded component)* plays a crucial role. Given a semi-Markovian graph $\mathcal{G}$ over a set of variables $\mathbf{V}$, there exists a unique partition such that each subgraph of $\mathcal{G}$ is a c-component.

**Definition 8** (C-component (Tian and Pearl, 2003))**.** Let $\mathcal{G}$ be a semi-Markovian graph such that a subset of its bidirected arcs forms a spanning tree over all vertices in $\mathcal{G}$. Then $\mathcal{G}$ is a *c-component*.

We denote by $\mathrm{cc}_{\mathcal{G}}$ the collection of maximal c-components so that $\mathrm{cc}_{\mathcal{G}} = \{\mathbf{C}_j\}_{j=1}^{l}$ implies that $\mathbf{C}_i$ is a maximal c-component, for each $\mathbf{C}_i \subseteq \mathbf{V}$, and there is no bidirected edge between $\mathbf{C}_i$ and $\mathbf{C}_j$ in $\mathcal{G}$ for $i \neq j$. Following Tian and Pearl (2003), for any $\mathbf{C} \subseteq \mathbf{V}$, we define function $Q[\mathbf{C}](\mathbf{v}) = P_{\mathbf{v} \setminus \mathbf{c}}(\mathbf{c})$. Moreover, $Q[\mathbf{V}](\mathbf{v}) = P(\mathbf{v})$ and $Q[\emptyset](\mathbf{v}) = 1$. For convenience, we omit input $\mathbf{v}$ and write $Q[\mathbf{C}]$. The importance of c-components lies in the following lemma.

**Lemma 1** (Lemma 3 in Tian and Pearl (2003))**.** *Let $\mathbf{C} \subseteq \mathbf{C}' \subseteq \mathbf{V}$, if $\mathbf{C}$ is an ancestral set in $\mathcal{G}[\mathbf{C}']$, then*

$$\sum_{\mathbf{c}' \setminus \mathbf{c}} Q[\mathbf{C}'] = Q[\mathbf{C}]. \tag{10}$$

**Lemma 2** (Lemma 4 in Tian and Pearl (2003))**.** *Let $\mathbf{C} \subseteq \mathbf{V}$, and assume that $\mathbf{C}$ is partitioned into c-components $\mathbf{C}_1, \cdots, \mathbf{C}_l$ in $\mathcal{G}[\mathbf{C}]$. Then,*

*(i)* $Q[\mathbf{C}]$ *can be decomposed as*

$$Q[\mathbf{C}] = \prod_{j=1}^{l} Q[\mathbf{C}_j] \tag{11}$$

*(ii)* *Let $\prec$ be a topological order over the variables in $\mathbf{C}$ according to $\mathcal{G}[\mathbf{C}]$ such that $C_1 \prec C_2 \cdots \prec C_k$. Let $\mathbf{C}^{\preceq i}$ be the variables in $\mathbf{C}$ that ordered before $C_i$ including $C_i$. Let $\mathbf{C}^{\succ i}$ be the variables in $\mathbf{C}$ that ordered after $C_i$. Then each $Q[\mathbf{C}_i]$ is computable from $Q[\mathbf{C}]$ and is given by*

$$Q[\mathbf{C}_j] = \prod_{C_i \in \mathbf{C}_j} \frac{Q[\mathbf{C}^{\preceq i}]}{Q[\mathbf{C}^{\preceq i-1}]} \tag{12}$$

*where each $Q[\mathbf{C}^{\preceq i}] = \sum_{\mathbf{C}^{\succ i}} Q[\mathbf{C}]$.*

*(iii)* *Each $\frac{Q[\mathbf{C}^{\preceq i}]}{Q[\mathbf{C}^{\preceq i-1}]}$ is a function only of $\mathbf{T}_i \triangleq \mathrm{Pa}(\bar{\mathbf{C}}_i)_{\mathcal{G}} \setminus \bar{\mathbf{C}}_i$, where $\bar{\mathbf{C}}_i$ is the c-component of $\mathcal{G}[\mathbf{C}^{\preceq i}]$ that contains $C_i$.*

Let $\mathbf{D}_1, \cdots, \mathbf{D}_k$ be the c-components of $\mathcal{G}$. Then, for any $\mathbf{C}_j \in \mathrm{cc}_{\mathcal{G} \setminus \mathbf{X}}$, there exists $\mathbf{C}_j \subseteq \mathbf{D}_i$ since if the variables in $\mathbf{C}_j$ are connected by a bidirected path in a subgraph of $\mathcal{G}$, they must also be connected in $\mathcal{G}$. Each c-factor $Q[\mathbf{C}_j]$ is identifiable if it is computable from $Q[\mathbf{D}_i]$, which can be determined recursively by repeatedly applying Lems 1 and 2. Based on this recursive strategy, Tian and Pearl (2003) proposed an identification algorithm that first decomposes the causal effect into a set of c-factors $Q[\mathbf{C}_j]$ and then checks the identifiability of each component iteratively.

**Bounding causal effect.** We now introduce the concept of *natural bounds* (Manski, 1990; Robins, 1989), which are functions of the observational data that consistently bound the causal effect $P_{\mathbf{x}}(\mathbf{y})$, regardless of the underlying causal structure of the system.

**Definition 9** (Natural bounds (Manski, 1990; Robins, 1989))**.** The natural bounds for a causal effect $P_{\mathbf{x}}(\mathbf{y})$ are given by

$$P(\mathbf{x}, \mathbf{y}) \leq P_{\mathbf{x}}(\mathbf{y}) \leq P(\mathbf{x}, \mathbf{y}) + 1 - P(\mathbf{x}). \tag{13}$$

## C  POSSIBLY OPTIMAL INTERVENTION SETS: CHARACTERIZATIONS

In this section, we provide graphical characterizations of POMIS and POIS, accompanied by illustrative examples. When given a causal diagram $\mathcal{G}$, *minimal unobserved confounders' territory* (MUCT) and *interventional border* (IB) provide a graphical characterization of PO(M)IS.

**Definition 10** (Unobserved-confounders' territory (Lee and Bareinboim, 2018))**.** Let $\mathcal{H} = \mathcal{G}[\mathrm{An}(Y)_{\mathcal{G}}]$. A set of variables $\mathbf{T} \subseteq \mathbf{V}(\mathcal{H})$ containing $Y$ is called a *UC-territory* on $\mathcal{G}$ with respect to $Y$ if $\mathrm{De}(\mathbf{T})_{\mathcal{H}} = \mathbf{T}$ and $\mathrm{CC}(\mathbf{T})_{\mathcal{H}} = \mathbf{T}$. If there is no $\mathbf{T}' \subsetneq \mathbf{T}$, we refer to it as a *minimal UC-territory* (MUCT) denoted as $\mathsf{MUCT}(\mathcal{G}, Y)$.

**Definition 11** (Interventional border (Lee and Bareinboim, 2018))**.** Let $\mathbf{T}$ be a minimal UC-territory on causal diagram $\mathcal{G}$ with respect to $Y$. Then $\mathrm{Pa}(\mathbf{T})_{\mathcal{G}} \setminus \mathbf{T}$ is called an *interventional border* (IB) for $\mathcal{G}$ with respect to $Y$ denoted as $\mathsf{IB}(\mathcal{G}, Y)$.

MUCT is the minimal set of variables that is closed under both descendants and bidirected connections; and IB consists of the parents of MUCT, excluding MUCT itself. Intuitively, MUCT is the minimal closed mechanism that conveys all hidden information from unobserved confounders to the downstream reward, while IB consists of the nodes that directly affect this closed mechanism.

**Theorem 4** (Theorem 6 in Lee and Bareinboim (2018))**.** *Given information $\langle \mathcal{G}, Y \rangle$, a set $\mathbf{X} \subseteq \mathbf{V} \setminus \{Y\}$ is a POMIS with respect to $\langle \mathcal{G}, Y \rangle$ if and only if $\mathsf{IB}(\mathcal{G}_{\overline{\mathbf{X}}}, Y) = \mathbf{X}$.*

Following the established structures, we provide a characterization for POIS with respect to $\langle \mathcal{G}, Y \rangle$.

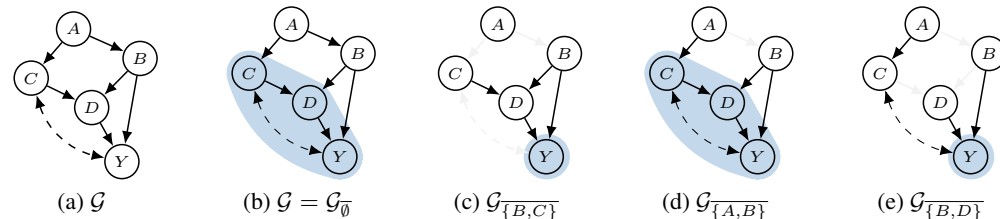

(a) $\mathcal{G}$    (b) $\mathcal{G} = \mathcal{G}_{\overline{\emptyset}}$    (c) $\mathcal{G}_{\overline{\{B,C\}}}$    (d) $\mathcal{G}_{\overline{\{A,B\}}}$    (e) $\mathcal{G}_{\overline{\{B,D\}}}$

Figure 8: The blue region illustrates MUCT. (b, c) are non-POMIS examples, while (d, e) correspond to POMIS. (e) Subsets $\{B, D\} \subseteq \mathbf{R} \subseteq \{A, B, C, D\}$ are POIS and share the same expected reward.

**Proposition 3** (Graphical characterization of POIS). *Let $\mathbf{T_X} \triangleq \mathsf{MUCT}(\mathcal{G}_{\overline{\mathbf{X}}}, Y)$. A set $\mathbf{X} \subseteq \mathbf{V} \setminus \{Y\}$ is a POIS with respect to $\langle \mathcal{G}, Y \rangle$ if and only if $\mathsf{IB}(\mathcal{G}_{\overline{\mathbf{X}}}, Y) \subseteq \mathbf{X} \subseteq \mathsf{An}(\mathbf{T_X})_{\mathcal{G}} \setminus \mathbf{T_X}$. Moreover, if $\mathbf{X} = \mathsf{IB}(\mathcal{G}_{\overline{\mathbf{X}}}, Y)$, then $\mathbf{X}$ is a POMIS with respect to $\langle \mathcal{G}, Y \rangle$.*

*Proof.* Note that $\mathsf{IB}(\mathcal{G}_{\overline{\mathbf{X}}}, Y) = \mathsf{Pa}(\mathbf{T_x})_{\mathcal{G}} \setminus \mathbf{T_X}$ is a POMIS with respect to $\langle \mathcal{G}, Y \rangle$ by Prop. 5 in (Lee and Bareinboim, 2018). If $\mathbf{X} = \mathsf{IB}(\mathcal{G}_{\overline{\mathbf{X}}}, Y)$, then $\mathbf{X}$ is a POMIS according to Thm. 4. Now, we consider $\mathsf{IB}(\mathcal{G}_{\overline{\mathbf{X}}}, Y) \subsetneq \mathbf{X} \subseteq \mathsf{An}(\mathbf{T_X})_{\mathcal{G}} \setminus \mathbf{T_X}$ We will show that there is no directed path from any $X \in \mathbf{X} \setminus \mathsf{IB}(\mathcal{G}_{\overline{\mathbf{X}}}, Y)$ to $Y$ that does not pass through $\mathsf{IB}(\mathcal{G}_{\overline{\mathbf{X}}}, Y)$, which implies $\mathbb{E}_{P_{\mathbf{x}}} Y = \mathbb{E}_{P_{\mathsf{IB}, \mathbf{x} \setminus \mathsf{IB}}} Y = \mathbb{E}_{P_{\mathsf{IB}}} Y = \mathbb{E}_{P_{\mathbf{x}[\mathsf{IB}]}} Y$. For the sake of contradiction, suppose that there exists a directed path from $X$ to $Y$ in $\mathcal{G}$ that does not pass through any node in $\mathsf{IB}(\mathcal{G}_{\overline{\mathbf{X}}}, Y)$. Since $Y \in \mathbf{T_X}$, its parent must belong to either $\mathbf{T_X}$ or $\mathsf{IB}(\mathcal{G}_{\overline{\mathbf{X}}}, Y)$. This implies that $X \in \mathbf{T_X}$ while also $X \in \mathsf{An}(\mathbf{T_X})_{\mathcal{G}} \setminus \mathbf{T_X}$, which leads to a contradiction. $\square$

For example, consider the causal diagram in Fig. 8e where $\{B, D\}$ is a POMIS with respect to $\langle \mathcal{G}, Y \rangle$ since $\mathsf{MUCT}(\mathcal{G}_{\overline{\{B,D\}}}, Y) = \{Y\}$ and $\mathsf{IB}(\mathcal{G}_{\overline{\{B,D\}}}, Y) = \{B, D\}$, which satisfies Thm. 4. Furthermore, $\{A, B, D\}$, $\{C, B, D\}$ and $\{A, B, C, D\}$ are POISs with respect to $\langle \mathcal{G}, Y \rangle$. Moreover, they are equivalent to the POMIS $\{B, D\}$.

**Corollary 2** (Equivalence). *Let $\mathbf{R} \subseteq \mathbf{V} \setminus \{Y\}$ be a POIS with respect to $\langle \mathcal{G}, Y \rangle$ and $\mathbf{R}^\dagger \triangleq \mathsf{IB}(\mathcal{G}_{\overline{\mathbf{R}}}, Y)$ denote the corresponding POMIS. Then, $\mathbf{R}$ and $\mathbf{R}^\dagger$ are equivalent in terms of expected reward.*

**Proposition 4** (Sharing transportability). *Let $\mathbf{R} \subseteq \mathbf{V} \setminus \{Y\}$ be a POIS with respect to $\langle \mathcal{G}, Y \rangle$. Then $\mathbf{R}^\dagger \triangleq \mathsf{IB}(\mathcal{G}_{\overline{\mathbf{R}}}, Y)$ is a POMIS with respect to $\langle \mathcal{G}, Y \rangle$. Moreover, they share a causal bound; given $\langle \mathcal{G}^{\boldsymbol{\Delta}}, \mathbb{Z} \rangle$, $\ell_{\mathbf{R}} = \ell_{\mathbf{R}^\dagger}$ and $u_{\mathbf{R}} = u_{\mathbf{R}^\dagger}$.*

*Proof.* This follows from the proof of Prop. 3. An important observation is that they share the same MUCT, $\mathsf{MUCT}(\mathcal{G}_{\overline{\mathbf{R}}}, Y)$, and that there is no path from $\mathbf{R}$ to $Y$ that does not pass through $\mathbf{R}^\dagger$. Therefore, $\mathbb{E}_{P^*_{\mathbf{r}[\mathbf{R}^\dagger]}} Y = \mathbb{E}_{P^*_{\mathbf{r}^\dagger}} Y \sum_y y P^*_{\mathbf{r}^\dagger}(y) = \sum_{\mathsf{MUCT}} y \prod_{q=1}^m Q^*[\mathbf{C}_q]$ (where $\mathbf{Y}^+ = \mathsf{MUCT}(\mathcal{G}_{\overline{\mathbf{R}}}, Y)$). The proof is thereby concluded by the application of Thm. 2. $\square$

**Proposition 5** (Theorem 4 in Lee and Bareinboim (2019)). *Given $\langle \mathcal{G}, Y, \mathbf{N} \rangle$, we have $\mathbb{P}^{\mathbf{N}}_{\mathcal{G}, Y} = \mathbb{P}_{\mathcal{H}, Y}$ where $\mathcal{H} = \mathcal{G} \langle \mathbf{V} \setminus \mathbf{N} \rangle$ is the latent projection of $\mathcal{G}$ onto $\mathbf{V} \setminus \mathbf{N}$.*

ALGORITHMIC CHARACTERIZATION OF POIS

The algorithm POISs (Alg. 2) is identical to POMISs (Alg. 1 in Lee and Bareinboim (2018)), except for Lines 3 and 9, where the set $\{\mathbf{X}\}$ is replaced with $\{\mathbf{R} \mid \mathbf{X} \subseteq \mathbf{R} \subseteq \mathsf{An}(\mathbf{T})_{\mathcal{G}} \setminus \mathbf{T}\}$ in order to include not only the POMIS $\mathbf{X}$ but also all POISs equivalent to it. The algorithm completely enumerates all POISs avoiding redundant computations by Thm. 9 in Lee and Bareinboim (2018) and it takes $\mathcal{O}(kn^2)$ where $k$ denotes the number of POIS and $n = |\mathbf{V}|$.

**Proposition 6.** *The algorithm POISs (Alg. 2) returns all, and only POISs given $\langle \mathcal{G}, Y \rangle$.*

*Proof.* This follows from Thm. 9 in Lee and Bareinboim (2018)) and Prop. 3. $\square$

---

**Algorithm 2:** Algorithm enumerating all POISs.

1 **function** POISs($\mathcal{G}, Y$)
2     **T**, **X** = MUCT($\mathcal{G}, Y$), IB($\mathcal{G}, Y$); $\mathcal{Q} = \mathcal{G}_{\overline{\mathbf{X}}}[\mathbf{T} \cup \mathbf{X}]$
3     **return** $\{\mathbf{R} \mid \mathbf{X} \subseteq \mathbf{R} \subseteq \mathrm{An}(\mathbf{T})_{\mathcal{G}} \setminus \mathbf{T}\} \cup$ subPOISs($\mathcal{Q}, Y$, reversed(topological-sort($\mathcal{Q}$)), $\emptyset$)
4 **function** subPOISs($\mathcal{G}, Y, \boldsymbol{\pi}, \mathbf{O}$)
5     $\mathbf{P} = \emptyset$
6     **for** $\pi[i] \in \boldsymbol{\pi}$ **do**
7         **T**, **X**, $\boldsymbol{\pi}'$, $\mathbf{O}'$ = MUCT($\mathcal{G}_{\overline{\pi[i]}}, Y$), IB($\mathcal{G}_{\overline{\pi[i]}}, Y$), $\boldsymbol{\pi}[i+1:]$, $\mathbf{O} \cup \boldsymbol{\pi}[1:i-1]$
8         **if** $\mathbf{X} \cap \mathbf{O}' = \emptyset$ **then**
9             $\mathbf{P} = \mathbf{P} \cup \{\mathbf{R} \mid \mathbf{X} \subseteq \mathbf{R} \subseteq \mathrm{An}(\mathbf{T})_{\mathcal{G}} \setminus \mathbf{T}\} \cup$ subPOISs($\mathcal{G}_{\overline{\mathbf{X}}}[\mathbf{T} \cup \mathbf{X}], Y, \boldsymbol{\pi}', \mathbf{O}'$) **if** $\boldsymbol{\pi}' \neq \emptyset$ **else** $\emptyset$.
10     **return** $\mathbf{P}$

---

## D   DOMINANCE BOUNDS

This section presents technical details relevant to the computation of dominance bounds.

### D.1   LOWER DOMINANCE BOUND

We start by demonstrating that the MISs (Alg. 3 in Lee and Bareinboim (2018)) is sound and complete under constraints.

**Lemma 3.** *The algorithm* MISs($\mathcal{G}\langle \mathbf{V} \setminus \mathbf{N} \rangle, Y$) *returns all and only MISs with respect to* $\langle \mathcal{G}, Y, \mathbf{N} \rangle$.

*Proof.* Let $\mathbf{X}$ be an MIS with respect to $\langle \mathcal{G}\langle \mathbf{V} \setminus \mathbf{N} \rangle, Y \rangle$. We will prove $\mathbf{X}$ is also an MIS with respect to $\langle \mathcal{G}, Y \rangle$ by proving the contrapositive. Suppose that $\mathbf{X}$ is *not* an MIS with respect to $\langle \mathcal{G}, Y \rangle$. Then, there exists a node $X \in \mathbf{X}$ such that there is no *proper directed path*[7] from $X$ to $Y$ with respect to $\mathbf{X}$ in $\mathcal{G}$. That is, every directed path from $X$ to $Y$ in $\mathcal{G}$ forms $X \to V_1 \to \cdots \to V_n \to Z \to W_1 \to \cdots \to W_m \to Y$ with $n, m \geq 0$ for an arbitrary $Z \in \mathbf{X} \setminus \{X\}$. Since the latent projection does not introduce any directed edges from $V_i$ to $W_j$ in $\mathcal{G}\langle \mathbf{V} \setminus \mathbf{N} \rangle$, all such paths correspond to non-proper directed paths from $X$ to $Y$ with respect to $\mathbf{X}$ in $\mathcal{G}\langle \mathbf{V} \setminus \mathbf{N} \rangle$. Therefore, $\mathbf{X}$ is *not* an MIS with respect to $\langle \mathcal{G}\langle \mathbf{V} \setminus \mathbf{N} \rangle, Y \rangle$, which completes the contrapositive. Now, suppose $\mathbf{X}$ is an MIS with respect to $\langle \mathcal{G}, Y \rangle$. By definition, there is no proper subset $\mathbf{X}' \subseteq \mathbf{V} \setminus \{Y\}$ such that $\mathbf{X}'$ is equivalent to $\mathbf{X}$. Moreover, since $\mathbf{X}$ is defined over $\mathbf{V} \setminus \{Y\} \setminus \mathbf{N}$, any such proper subset $\mathbf{X}'$ must also be defined over the same superset. Hence, any set $\mathbf{X}$ included in the output of MISs($\mathcal{G}\langle \mathbf{V} \setminus \mathbf{N} \rangle, Y$) is an MIS with respect to $\langle \mathcal{G}, Y, \mathbf{N} \rangle$. Combining this with the soundness and completeness of the MISs algorithm, the proof is complete. $\square$

Equipped with this result and the definition of MIS (Def. 4), the lower dominance bound $\ell^\star$ can be computed using MISs; $\ell^\star = \max_{\mathbf{w} \in \Omega_\mathbf{W}, \mathbf{W} \in \mathrm{MISs}(\mathcal{G}\langle \mathbf{V} \setminus \mathbf{N}^* \rangle, Y)}$.

### D.2   UPPER DOMINANCE BOUND

We proceed to the upper dominance bound. It is crucial to identify a space in which POMISs under different constraints can be meaningfully compared. The following lemma shows that POMISs remain MISs under weaker constraints.

**Lemma 4.** *If* $\mathbf{X}$ *is a (PO)MIS with respect to* $\langle \mathcal{G}, Y, \mathbf{N} \rangle$, *then it is also an MIS with respect to* $\langle \mathcal{G}, Y \rangle$.

*Proof.* In the proof of Lem. 3, we have shown that if $\mathbf{X}$ be an MIS with respect to $\langle \mathcal{G}\langle \mathbf{V} \setminus \mathbf{N} \rangle, Y \rangle$, then it is also an MIS with respect to $\langle \mathcal{G}, Y \rangle$. Since the set of POMISs is a subset of the set of MISs under the same constraint, the result also holds for the POMISs, which concludes the proof. $\square$

**Lemma 5.** *Let* $\mathbb{P}_{\mathcal{G}, Y}^{\mathbf{N}'}$ *and* $\mathbb{P}_{\mathcal{G}, Y}^{\mathbf{N}}$ *be a set of POMISs corresponding to the constraints* $\mathbf{N}'$ *and* $\mathbf{N}$, *respectively, where* $\mathbf{N}' \subseteq \mathbf{N}$. *Then* $\mathbb{P}_{\mathcal{G}, Y}^{\mathbf{N}}$ *dominates* $\mathbb{P}_{\mathcal{G}, Y}^{\mathbf{N}'}$.

---

[7]We refer to a directed path from $X \in \mathbf{X}$ to $Y$ as a *proper* directed path with respect to $\mathbf{X}$ if only the first node $X$ belongs to $\mathbf{X}$.

*Proof.* Let $\mathcal{H} \triangleq \mathcal{G}\langle \mathbf{V} \setminus \mathbf{N}' \rangle$. According to Prop. 5, we have $\mathbb{P}_{\mathcal{G},Y}^{\mathbf{N}'} = \mathbb{P}_{\mathcal{H},Y}^{\mathbf{N} \setminus \mathbf{N}'}$ and $\mathbb{P}_{\mathcal{G},Y}^{\mathbf{N}} = \mathbb{P}_{\mathcal{H},Y}$. Let us denote $\mathbf{N}^{\dagger} = \mathbf{N} \setminus \mathbf{N}'$. According to Lem. 4, any $\mathbf{X} \in \mathbb{P}_{\mathcal{H},Y}^{\mathbf{N}^{\dagger}}$ is an MIS with respect to $\langle \mathcal{H}, Y \rangle$. Since POMISs dominate MISs under the same constraint $\mathbf{N}^{\dagger}$, the claim follows. $\qquad\square$

**Theorem 1** (Dominance relationship). *Let $\mathbf{r}^{\star}$ be an optimal action with respect to $\langle \mathcal{G}, Y, \mathbf{N} \rangle$ where $\mathbf{N}$ is a subset of $\mathbf{N}^*$. Let $\mathbf{W}$ be a non-POIS with respect to $\langle \mathcal{G}, Y, \mathbf{N}^* \rangle$. Then $\mathbb{E}_{P_{\mathbf{x}^{\star}}^*} Y$ is bounded by*

$$\mathbb{E}_{P_{\mathbf{w}^{\star}}^*} Y \leq \mathbb{E}_{P_{\mathbf{x}^{\star}}^*} Y \leq \mathbb{E}_{P_{\mathbf{r}^{\star}}^*} Y. \tag{4}$$

*Proof.* Without loss of generality, we assume $\mathbf{N} = \emptyset$ since we can equivalently reformulate the graph as $\mathcal{G} = \mathcal{G}\langle \mathbf{V} \setminus \mathbf{N} \rangle$ and the constraint set as $\mathbf{N}^* = \mathbf{N}^* \setminus \mathbf{N}$. According to Lem. 4, any $\mathbf{X} \in \mathbb{P}_{\mathcal{G},Y}^{\mathbf{N}^*}$ is an MIS with respect to $\langle \mathcal{G}, Y \rangle$. Furthermore, for any unconstrained POISs $\mathbf{R}$, we can always find an equivalent unconstrained POMIS $\mathbf{R}^{\dagger} = \mathsf{IB}(\mathcal{G}_{\overline{\mathbf{R}}}, Y)$ that yields the same expected reward, i.e., $\mathbb{E}_{P_{\mathbf{r}}} Y = \mathbb{E}_{P_{\mathbf{r}[\mathbf{R}^{\dagger}]}} Y$, as supported by Cor. 2. Therefore, the upper dominance bound is sound. The lower bound also holds by the definition of MIS (Def. 4) and POIS (Def. 5). $\qquad\square$

Building on the statements, we present the algorithm UDB (Alg. 3) which returns a valid upper dominance bound $u^{\star}$ given the inputs $(\mathcal{G}, Y, \mathbf{N}, \mathbb{U})$ where $\mathbb{U}$ denotes a collection of upper causal bounds for all actions.

---

**Algorithm 3:** Upper Dominance Bound (UDB)

---

1 **function** $\text{UDB}(\mathcal{G}, Y, \mathbf{N}, \mathbb{U})$:
  **Input:** $\mathcal{G}$: causal diagram; $Y$: reward variable; $\mathbf{N}$: non-manipulable variables;
       $\mathbb{U} = \{u_{\mathbf{x}} \mid \mathbf{x} \in \Omega_{\mathbf{X}}, \mathbf{X} \in 2^{\mathbf{V} \setminus \{Y\}}\}$: collection of upper causal bounds.
1 **Ensure:** All upper causal bounds $u_{\mathbf{x}} \in \mathbb{U}$ have been computed.
2      Compute the latent projection $\mathcal{H} = \mathcal{G}\langle \mathbf{V} \setminus \mathbf{N} \rangle$; Initialize the upper dominance bound $u^{\star} = 0$.
3      **for** $\mathbf{R} := \mathsf{POMISs}(\mathcal{H}, Y)$ **do if** $u^{\star} > \max_{\mathbf{r} \in \Omega_{\mathbf{R}}} u_{\mathbf{r}}$ **then** $(u^{\star}, \mathbf{R}^{\star}) = (\max_{\mathbf{r} \in \Omega_{\mathbf{R}}} u_{\mathbf{r}}, \mathbf{R})$.
4      **if** $u^{\star} < \infty$ and $u_{\mathbf{r}^{\star}} = \ell_{\mathbf{r}^{\star}}$ (i.e., transportable) **then return** $u^{\star}$
5      **if** $\mathbf{N} = \emptyset$ **then return** $u^{\star}$
6      **for all** $\mathbf{N}' \subset \mathbf{N}$ such that $|\mathbf{N}'| = |\mathbf{N}| - 1$ **do** $u^{\star} = \min\{u^{\star}, \text{UDB}(\mathcal{G}, Y, \mathbf{N}', \mathbb{U})\}$
7      **return** $u^{\star}$

---

In Line 3, the algorithm attempts to compute upper causal bounds of POMISs. If the algorithm reaches Line 6, this implies that (1) there exists at least one POMIS action whose upper causal bound is $\infty$, or (2) the current upper dominance bound corresponds to upper causal bound of a *non*-transportable action. In such cases, the algorithm recursively explores weaker constraints $\mathbf{N}'$ and returns the tightest bound among the results of recursive calls; on the other hand, if $u^{\star} < \infty$ and the upper dominance bound corresponds to the expected reward of a transportable action (Line 4), no further recursive calls are required—this is justified by Lem. 5—and the algorithm simply returns $u^{\star}$.

**Runtime analysis.** In the worst case, the algorithm may need to traverse up to the unconstrained POMISs, resulting in an exponential time complexity in the size of $\mathbf{N}^*$, i.e., $\mathcal{O}(2^{|\mathbf{N}^*|})$.

**Dominance relationship example.** As a concrete example to illustrate the dominance relationships in Fig. 3, we present an SCM that exhibits these relations. We consider an SCM $\mathcal{M}$ where

$$\mathcal{M} = \begin{cases} \mathbf{U} & = \{U_A, U_B, U_C, U_Y, U_{BC}, U_{BY}\} \\ \mathbf{V} & = \{A, B, C, Y\} \\ \mathbf{F} & = \begin{cases} f_A = u_{AB}, f_B = u_B \oplus u_{BC} \oplus u_{BY}, \\ f_C = a \oplus b \wedge u_C, f_Y = u_A \oplus u_C \oplus u_Y \wedge u_{BY} \end{cases} \\ P(\mathbf{U}) & = \begin{cases} U_A \sim \texttt{Bern}(0.3), U_B \sim \texttt{Bern}(0.1), U_C \sim \texttt{Bern}(0.25), \\ U_Y \sim \texttt{Bern}(0.2), U_{BC} \sim \texttt{Bern}(0.2), U_{BY} \sim \texttt{Bern}(0.15). \end{cases} \end{cases} \tag{14}$$

To elaborate on dominance relations, let us suppose access to the causal bounds of expected rewards of all actions. In this setting, the value of upper dominance bound is:

(d) $u^\star = \max_{\mathbf{r}\in\Omega_{\mathbf{R}},\mathbf{R}\in\mathbb{P}_{\mathcal{G},Y}^{\{A,C\}}=\{\emptyset,\{B\}\}} \mathbb{E}_{P_{\mathbf{r}}^*} Y \approx 0.265$. For weaker constraints, we have (b) $\max_{\mathbf{r}\in\Omega_{\mathbf{R}},\mathbf{R}\in\mathbb{P}_{\mathcal{G},Y}^{\{A\}}=\{\emptyset,\{B\},\{C\}\}} \mathbb{E}_{P_{\mathbf{r}}^*} Y \approx 0.782$, (c) $\max_{\mathbf{r}\in\Omega_{\mathbf{R}},\mathbf{R}\in\mathbb{P}_{\mathcal{G},Y}^{\{C\}}=\{\emptyset,\{A\},\{A,B\}\}} \mathbb{E}_{P_{\mathbf{r}}^*} Y \approx 0.782$; and (a) $\max_{\mathbf{r}\in\Omega_{\mathbf{R}},\mathbf{R}^\dagger\in\mathbb{P}_{\mathcal{G},Y}=\{\emptyset,\{A\},\{A,C\}\}} \mathbb{E}_{P_{\mathbf{r}}^*} Y \approx 0.97$.

Therefore, we observe the dominance relationships: (d) $\leq$ (b, c) $\leq$ (a).

# E    CAUSAL BOUNDS: PARTIAL TRANSPORTABILITY

This section presents technical details relevant to the computation of causal bounds (Prop. 1 and thm. 2 in Sec. 4). We begin by introducing counterfactual variables, which play a useful role in our proofs.

**Counterfactual variables.**    Given a set of variables $\mathbf{Y} \subseteq \mathbf{V}$, the solution for $\mathbf{Y}$ in $\mathcal{M}_{\mathbf{x}}$ defines a *potential response* for a unit $\mathbf{u}$, denoted as $\mathbf{Y}_{\mathbf{x}}(\mathbf{u})$. Averaging over the space of $\mathbf{U}$, a potential response $\mathbf{Y}_{\mathbf{x}}(\mathbf{u})$ induces a *counterfactual variables* $\mathbf{Y}_{\mathbf{x}}$ (Pearl, 2000).

Correa and Bareinboim (2025) introduced a novel calculus over probability quantities may defined at the counterfactual level, called the *ctf-calculus*. The independence rule (Rule 2) in ctf-calculus requires the construction of another graphical object, known as the *Ancestral Multi-World Network* (AMWN), which serves to identify d-separation (Pearl, 1995) relations among counterfactual variables.

**Theorem 5** (Counterfactual calculus (ctf-calculus); Theorem 3.1 in Correa and Bareinboim (2025))**.**
*Let $\mathcal{G}$ be a causal diagram, then for $\mathbf{Y}$, $\mathbf{X}$, $\mathbf{Z}$, $\mathbf{W}$, $\mathbf{T}$, $\mathbf{R} \subseteq \mathbf{V}$, the following rules hold for the probability distributions generated by any model compatible with $\mathcal{G}$:*

> *Rule 1. (Consistency rule - Obs./intervention exchange)*
> $P(\mathbf{y}_{\mathbf{T}_*\mathbf{x}}, \mathbf{x}_{\mathbf{T}_*}, \mathbf{w}_*) = P(\mathbf{y}_{\mathbf{T}_*}, \mathbf{x}_{\mathbf{T}_*}, \mathbf{w}_*)$

> *Rule 2. (Independence rule - Adding/removing counterfactual observations)*
> $P(\mathbf{y}_{\mathbf{r}} \mid \mathbf{x}_{\mathbf{t}}, \mathbf{w}_*) = P(\mathbf{y}_{\mathbf{r}}, \mathbf{w}_*)$
> $\quad if\ (\mathbf{Y}_{\mathbf{r}} \perp\!\!\!\perp \mathbf{X}_{\mathbf{t}} \mid \mathbf{W}_*)_{\mathcal{G}_A}$

> *Rule 3. (Exclusion Rule - Adding/removing interventions)*
> $P(\mathbf{y}_{\mathbf{xz}}, \mathbf{w}_*) = P(\mathbf{y}_{\mathbf{z}}, \mathbf{w}_*)$
> $\quad if\ \mathbf{X} \cap \mathtt{An}(\mathbf{Y}) = \emptyset\ in\ \mathcal{G}_{\overline{\mathbf{Z}}}$

*where $\mathcal{G}_A$ is the AMWN $\mathcal{G}_A(\mathcal{G}, \mathbf{Y}_{\mathbf{r}} \cup \mathbf{X}_{\mathbf{t}} \cup \mathbf{W}_*)$.*

However, since only Rule 1 (**R1**) is used in this paper, we refer the reader to Correa and Bareinboim (2025) for further details about AMWN.

**Proposition 1.** *Let $\mathbf{C}$ be a c-component in $\mathcal{G} \setminus \mathbf{X}$ satisfying $\mathbf{C} \cap \Delta^i = \emptyset$ and $\mathbf{D}$ be a c-component satisfying $\mathbf{C} \subseteq \mathbf{D}$. Let $\mathbf{C}' = \mathtt{An}(\mathbf{C})_{\mathcal{G}[\mathbf{D}]}$. The target c-factor $Q^*[\mathbf{C}]$ is bounded in $[\ell, u]$ where (i) if $\mathbf{C} = \mathbf{C}'$, then $\ell = u = \sum_{\mathbf{c}'\setminus\mathbf{c}} Q^i[\mathbf{C}']$; (ii) otherwise, $\ell = Q^i[\mathbf{C}']$ and $u = Q^i[\mathbf{C}'] + 1 - \sum_{\mathbf{c}} Q^i[\mathbf{C}']$.*

*Proof.* We prove ours build on the notion of counterfactual variables. We use $P(\mathbf{y}_{\mathbf{x}})$ for probabilities $P(\mathbf{Y}_{\mathbf{x}} = \mathbf{y})$. By definition, $P_{\mathbf{x}}(\mathbf{y}) = P(\mathbf{y}_{\mathbf{x}})$. According to Lem. 1, $\mathtt{An}(\mathbf{C})_{\mathcal{G}[\mathbf{C}']} = \mathbf{C}$ implies $Q^i[\mathbf{C}] = \sum_{\mathbf{c}'\setminus\mathbf{c}} Q^i[\mathbf{C}']$. Further, we have that

$$Q^*[\mathbf{C}] = P_{\mathbf{v}\setminus\mathbf{c}}^{\mathbf{\Delta}}(\mathbf{c} \mid \mathbf{s}^{\mathbf{i}} = \mathbf{0}, \mathbf{s}^{-\mathbf{i}} = \mathbf{0}) = P_{\mathbf{v}\setminus\mathbf{c}}^{\mathbf{\Delta}}(\mathbf{c} \mid \mathbf{s}^{\mathbf{i}} = \mathbf{i}, \mathbf{s}^{-\mathbf{i}} = \mathbf{0}) = Q^i[\mathbf{C}]$$

holds by $\mathbf{C} \cap \Delta^i = \emptyset$, and thus we have $Q^*[\mathbf{C}] = \sum_{\mathbf{c}'\setminus\mathbf{c}} Q^i[\mathbf{C}']$. Otherwise $\mathtt{An}(\mathbf{C})_{\mathcal{G}[\mathbf{C}']} \neq \mathbf{C}$, without loss of generality, we have $\mathtt{An}(\mathbf{C})_{\mathcal{G}[\mathbf{C}']} = \mathbf{C}'$. Let $\mathbf{T} \triangleq \mathbf{C}' \setminus \mathbf{C}$, i.e., $\mathbf{C}' = \mathbf{T} \cup \mathbf{C}$. By basic probabilistic algebra,

$$Q^*[\mathbf{C}] = Q^i[\mathbf{C}] = P^i(\mathbf{c}_{\mathbf{v}\setminus\mathbf{c}}) = \sum_{\mathbf{t}'} P^i(\mathbf{c}_{\mathbf{v}\setminus\mathbf{c}}, \mathbf{t}'_{\mathbf{v}\setminus\mathbf{c}'}) = \sum_{\mathbf{t}'} P^i(\mathbf{c}_{\mathbf{v}\setminus\mathbf{c}'}, \mathbf{t}'_{\mathbf{v}\setminus\mathbf{c}'}) \tag{15}$$

$$\geq P^i(\mathbf{c}_{\mathbf{v}\setminus\mathbf{c}',\mathbf{t}}, \mathbf{t}_{\mathbf{v}\setminus\mathbf{c}'}) \overset{\mathbf{R1}}{=} P^i(\mathbf{c}_{\mathbf{v}\setminus\mathbf{c}'}, \mathbf{t}_{\mathbf{v}\setminus\mathbf{c}'}) = Q^i[\mathbf{C}']. \tag{16}$$

We thus have $Q^*[\mathbf{C}] \geq Q^i[\mathbf{C}']$. Similarly, we prove the upper bound of $Q^*[\mathbf{C}]$. Using basic probabilistic algebra,

$$Q^*[\mathbf{C}] = Q^i[\mathbf{C}] = P^i(\mathbf{c_{v\setminus c}}) \tag{17}$$

$$= \sum_{\mathbf{t}'} P^i(\mathbf{c_{v\setminus c}}, \mathbf{t}'_{\mathbf{v\setminus c'}}) \tag{18}$$

$$\overset{\mathbf{R1}}{=} P^i(\mathbf{c_{v\setminus c'},t}, \mathbf{t_{v\setminus c'}}) + \sum_{\mathbf{t}'\neq\mathbf{t}} P^i(\mathbf{c_{v\setminus c'},t}, \mathbf{t}'_{\mathbf{v\setminus c'}}) \tag{19}$$

$$\leq P^i(\mathbf{c_{v\setminus c'},t}, \mathbf{t_{v\setminus c'}}) + \sum_{\mathbf{t}'\neq\mathbf{t}} P^i(\mathbf{t}'_{\mathbf{v\setminus c'}}) \tag{20}$$

$$\overset{\mathbf{R1}}{=} P^i(\mathbf{c_{v\setminus c'}}, \mathbf{t_{v\setminus c'}}) + 1 - P^i(\mathbf{t_{v\setminus c'}}) \tag{21}$$

$$= P^i(\mathbf{c_{v\setminus c'}}, \mathbf{t_{v\setminus c'}}) + 1 - \sum_{\mathbf{c}} P^i(\mathbf{c_{v\setminus c'}}, \mathbf{t_{v\setminus c'}}) \tag{22}$$

$$= Q^i[\mathbf{C}'] + 1 - \sum_{\mathbf{c}} Q^i[\mathbf{C}']. \tag{23}$$

We thus have $Q^*[\mathbf{C}] \leq Q^i[\mathbf{C}'] + 1 - \sum_{\mathbf{c}} Q^i[\mathbf{C}']$ which concludes the proof. $\square$

**Theorem 2** (Causal bounds). *Given $\langle \mathcal{G}^{\mathbf{\Delta}}, \mathbb{Z} \rangle$, the target expected reward $\mathbb{E}_{P^*_{\mathbf{x}}} Y$ can be bounded by $[\ell_{\mathbf{x}}, u_{\mathbf{x}}]$ if for all c-factors $Q^*[\mathbf{C}_q]$ in the right-hand side of Eq. (6), there exists a source $\pi^i \in \Pi$ satisfying $\mathbf{C}_q \cap \Delta^i = \emptyset$ and a computable ancestral c-component $\mathbf{C}_q \subseteq \mathbf{C}'_q$ from $\mathcal{G} \setminus \mathbf{Z}$ where $\mathbf{Z} \in \mathbb{Z}^i \in \mathbb{Z}$. The bound $[\ell_{\mathbf{x}}, u_{\mathbf{x}}]$ is defined as:*

$$\ell_{\mathbf{x}} \triangleq \sum_{\mathbf{y}^+} y \prod_{q=1}^{k} Q^{i_q}[\mathbf{C}_q] \prod_{q=k+1}^{m} Q^{j_q}[\mathbf{C}'_q] \leq \sum_{\mathbf{y}^+} y \prod_{q=1}^{k} Q^{i_q}[\mathbf{C}_q] \prod_{q=k+1}^{m} Q^*[\mathbf{C}_q] \tag{7}$$

$$\leq \sum_{y} y \min\Big\{1, \sum_{\mathbf{y}^+\setminus\{y\}} \prod_{q=1}^{k} Q^{i_q}[\mathbf{C}_q] \prod_{q=k+1}^{m} \big\{ Q^{j_q}[\mathbf{C}'_q] + 1 - \sum_{\mathbf{c}_q} Q^{j_q}[\mathbf{C}'_q] \big\} \Big\} \triangleq u_{\mathbf{x}} \tag{8}$$

*where $\prod_{q=1}^{k} Q^{i_q}[\mathbf{C}_q]$ denotes transportable c-factors.*

*Proof.* Recall Eq. (6), $\mathbb{E}_{P^*_{\mathbf{x}}} Y = \sum_y y P^*_{\mathbf{x}}(y) = \sum_{\mathbf{y}^+} y \prod_{q=1}^{m} Q^*[\mathbf{C}_q]$. This expression can be decomposed into *transportable* and *non-transportable* components as follows: $\mathbb{E}_{P^*_{\mathbf{x}}} Y = \sum_{\mathbf{y}^+} y \prod_{q=1}^{k} Q^{i_q}[\mathbf{C}_q] \prod_{q=k+1}^{m} Q^*[\mathbf{C}_q]$ where the first product represents the transportable terms. According to Prop. 1, each non-transportable c-factor $Q^*[\mathbf{C}_q]$ satisfies the inequality $Q^i[\mathbf{C}'_q] \leq Q^*[\mathbf{C}_q] \leq Q^i[\mathbf{C}'_q] + 1 - \sum_{\mathbf{c}_q} Q^i[\mathbf{C}'_q]$. Therefore, by substituting this inequality into the non-transportable components of the decomposition. Moreover, note that both Eq. (6) and Eqs. (7) and (8) are functions of $\mathbf{X} \cup \mathbf{Y}^+$ where $\mathbf{Y}^+ = \text{An}(Y)_{\mathcal{G}_{\mathbf{X}}}$. Therefore, the expressions for $\ell_{\mathbf{x}}$ and $u_{\mathbf{x}}$ are valid. Moreover, taking the maximum or minimum over $\mathbf{Y}^+$, with respect to $\mathcal{P}^{\Pi}_{\mathbb{Z}}$, returns valid bounds, equipped with $\min\{1, \cdot\}$. $\square$

### E.1 Partial Transportability of Causal Effects

**Definition 12** (Partial-transportability). Let $\mathcal{G}^{\mathbf{\Delta}}$ be a collective selection diagram with respect to $\Pi = \{\pi_1, \cdots, \pi_n\}$ with a target domain $\pi^*$. Let $\mathbb{Z} = \{\mathbb{Z}^i\}_{i=1}^n$ be a specification of actions $\mathbb{Z}^i$ conducted in source environment $\pi^i$. We say that $P^*_{\mathbf{x}}(y)$ is *partially transportable* with respect to $\langle \mathcal{G}^{\mathbf{\Delta}}, \mathbb{Z} \rangle$ if it determines a bound $[\ell, u]$ for $P^*_{\mathbf{x}}(y)$ that is strictly contained in $[0, 1]$ and valid over $\mathcal{P}^{\Pi}_{\mathbb{Z}} = \{ P^i_{\mathbf{z}} \mid \mathbf{z} \in \Omega_{\mathbf{Z}}, \mathbf{Z} \in \mathbb{Z}^i \in \mathbb{Z} \}$ in any collection of models that induce $\mathcal{G}^{\mathbf{\Delta}}$.

We propose the partial-transportability algorithm PATR (Alg. 4) which returns expressions of bounds for $P^*_{\mathbf{x}}(y)$. The prior specification $\mathbb{Z}$ and the corresponding distributions $\mathcal{P}^{\Pi}_{\mathbb{Z}}$ are defined globally and do not change with the specific invocation of the algorithm. In contrast, variables $\mathbf{V}$ and selection variables $\mathbf{S}$ reflect graph $\mathcal{G}$ and discrepancies $\mathbf{\Delta}$, respectively, relative to the arguments passed to the current execution of the procedure.

In the algorithm, Line 5 breaks down the query into queries where $\mathbf{Y}$ in each sub-query forms a c-component. Line 6 examines whether some experimental distribution $P^i_{\mathbf{z}} \in \mathcal{P}^{\Pi}_{\mathbb{Z}}$ can be used to

---

**Algorithm 4:** Partial-transportability algorithm (PATR).

1 **function** PATR$(\mathbf{y}, \mathbf{x}, \mathcal{G}, \boldsymbol{\Delta}, \mathsf{type})$
   **Input:** $\mathbf{y}$; $\mathbf{x}$; $\mathcal{G}$: causal diagram; $\boldsymbol{\Delta}$: discrepancies; $\mathsf{type} \in \{\ell, u\}$: type of bound.

2   **if** $\exists \mathbf{Z} \in \mathbb{Z}^i \in \mathbb{Z}$ such that $(\mathbf{X} = \mathbf{Z} \cap \mathbf{V}) \wedge (\mathbf{S}^i \perp\!\!\!\perp_d \mathbf{Y})_{\mathcal{G}\boldsymbol{\Delta} \setminus \mathbf{x}}$ **then yield** $P^i_{\mathbf{z} \setminus \mathbf{V}, \mathbf{x} \cap \mathbf{z}}(\mathbf{y})$.

3   **if** $\mathbf{V}' := \mathbf{V} \setminus \mathtt{An}(\mathbf{Y})_{\mathcal{G}} \neq \emptyset$ **then yield** PATR$(\mathbf{y}, \mathbf{x} \setminus \mathbf{V}', \mathcal{G} \setminus \mathbf{V}', \{\Delta^i \setminus \mathbf{V}' \mid \Delta^i \in \boldsymbol{\Delta}\})$.

4   **if** $\mathbf{V}' := (\mathbf{V} \setminus \mathbf{X}) \setminus \mathtt{An}(\mathbf{Y})_{\mathcal{G}_{\overline{\mathbf{X}}}} \neq \emptyset$ **then yield** PATR$(\mathbf{y}, \mathbf{x} \cup \mathbf{V}', \mathcal{G}, \boldsymbol{\Delta})$.

5   **if** $|\mathsf{cc}_{\mathcal{G} \setminus \mathbf{x}}| > 1$ **then yield** $\sum_{\mathbf{v} \setminus (\mathbf{y} \cup \mathbf{x})} \prod_{\mathbf{C} \in \mathsf{cc}_{\mathcal{G} \setminus \mathbf{x}}}$ PATR$(\mathbf{c}, \mathbf{v} \setminus \mathbf{c}, \mathcal{G}, \boldsymbol{\Delta})$.

6   **for** $\pi^i \in \Pi$ such that $\Delta^i \cap (\mathbf{V} \setminus \mathbf{X}) = \emptyset$, **for** $\mathbf{Z} \in \mathbb{Z}^i$ such that $\mathbf{Z} \cap \mathbf{V} \subseteq \mathbf{X}$ **do**

7      **yield** PAID$(\mathbf{y}, \mathbf{x} \setminus \mathbf{Z}, P^i_{\mathbf{z} \setminus \mathbf{V}, \mathbf{x} \cap \mathbf{Z}}, \mathcal{G} \setminus (\mathbf{Z} \cap \mathbf{X}), \mathsf{type})$

8 **function** PAID$(\mathbf{y}, \mathbf{x}, P, \mathcal{G}, \mathsf{type})$

9   $\{\mathbf{C}\} \leftarrow \mathsf{cc}_{\mathcal{G} \setminus \mathbf{x}}$.

10   **if** $\mathbf{X} = \emptyset$ **then yield** $\sum_{\mathbf{v} \setminus \mathbf{y}} P(\mathbf{v})$.

11   **if** $\mathbf{V} \neq \mathtt{An}(\mathbf{Y})_{\mathcal{G}}$ **then yield** PAID$(\mathbf{y}, \mathbf{x} \cap \mathtt{An}(\mathbf{Y})_{\mathcal{G}}, \sum_{\mathbf{v} \setminus \mathtt{An}(\mathbf{Y})_{\mathcal{G}}} P(\mathbf{v}), \mathcal{G}[\mathtt{An}(\mathbf{Y})_{\mathcal{G}}], \mathsf{type})$.

12   **if** $\mathsf{cc}_{\mathcal{G}} = \{\mathbf{V}\}$ **then; if** $\mathsf{type} = \ell$ **then yield** $\sum_{\mathbf{v} \setminus (\mathbf{y} \cup \mathbf{x})} P(\mathbf{v})$ **else yield**
       $\sum_{\mathbf{v} \setminus (\mathbf{y} \cup \mathbf{x})} \{P(\mathbf{v}) + 1 - \sum_{\mathbf{v} \setminus \mathbf{x}} P(\mathbf{v})\}$.

13   **if** $\mathbf{C} \in \mathsf{cc}_{\mathcal{G}}$ **then** $\sum_{\mathbf{C} \setminus \mathbf{y}} \prod_{V_i \in \mathbf{Y}} P(v_i \mid \mathbf{v}^{(i-1)}_{\pi})$.

14   **if** $\mathbf{C} \subsetneq \mathbf{C}' \in \mathsf{cc}_{\mathcal{G}}$ **then yield**
       PAID$(\mathbf{y}, \mathbf{x} \cap \mathbf{C}', \prod_{V_i \in \mathbf{C}'} P(V_i \mid \mathbf{V}^{(i-1)}_{\pi} \cap \mathbf{C}', \mathbf{v}^{(i-1)}_{\pi} \setminus \mathbf{C}'), \mathcal{G}[\mathbf{C}'], \mathsf{type})$.

---

identify the query. If valid, PATR passes the query to PAID with a slight modification of it and graph, taking into account the shared intervention between $\mathbf{Z}$ and $\mathbf{X}$.

*Remark.* When the subroutine PAID is called, the following conditions are satisfied (related to Lines 4 and 5): $(\mathbf{V} \setminus \mathbf{X}) \setminus \mathtt{An}(\mathbf{Y})_{\mathcal{G}_{\overline{\mathbf{X}}}} = \emptyset$ and $\mathcal{G} \setminus \mathbf{X}$ is a c-component.

PATR enumerates *all* expressions through the subroutine PAID, which returns lower or upper bound depending on input $\mathsf{type} \in \{\ell, u\}$. The condition of Thm. 2 is inherently captured within PATR.

**Proposition 7** (Soundness). PATR *(Alg. 4) returns all expressions of upper (lower) bounds of $P^*_{\mathbf{x}}(y)$.*

*Proof.* Let a superscript $l$ denote variables and values local to the function. The soundness of the algorithm was partially proved by Lee et al. (2020) under the case where the given query is transportable. Our main interest is the case where $P^*_{\mathbf{x}}(y)$ is non-transportable but holds the condition in Prop. 1 is satisfied. First, we show that $Q^i[\mathbf{V}_l] \leq P^*_{\mathbf{x}_l}(\mathbf{v}_l \setminus \mathbf{x}_l) \leq Q^i[\mathbf{V}_l] + 1 - \sum_{\mathbf{v}_l \setminus \mathbf{x}_l} Q^i[\mathbf{V}_l]$ at Line 12. Note that $P^*_{\mathbf{x}_l}(\mathbf{v}_l \setminus \mathbf{x}_l) = P^i_{\mathbf{x}_l}(\mathbf{v}_l \setminus \mathbf{x}_l)$ holds by Lem. 1 in Lee et al. (2020). Moreover, it holds that $\mathbf{V}_l = \mathtt{An}(\mathbf{Y}_l)_{\mathcal{G}_l}$ and $\mathbf{V}_l$ is a c-component in $\mathcal{G}_l$, as ensured by earlier steps in PAID. Therefore, we have $Q[\mathbf{V}_l] = P^i(\mathbf{v}_l) \leq P^*_{\mathbf{x}_l}(\mathbf{v}_l \setminus \mathbf{x}_l) \leq P^i(\mathbf{v}_l) + 1 - \sum_{\mathbf{v}_l \setminus \mathbf{x}_l} P^i(\mathbf{v}_l) = Q^i[\mathbf{V}_l] + 1 - \sum_{\mathbf{v}_l \setminus \mathbf{x}_l} Q^i[\mathbf{V}_l]$ due to Prop. 1. Taking $P^*_{\mathbf{x}_l}(\mathbf{y}_l) = \sum_{\mathbf{v}_l \setminus (\mathbf{x}_l \cup \mathbf{y}_l)} P^*_{\mathbf{x}_l}(\mathbf{v}_l \setminus \mathbf{x}_l)$ concludes the proof. $\square$

Any value of PATR$(\mathsf{type} = \ell)$ is always less than or equal to $P^*_{\mathbf{x}}(y)$. However, for PATR$(\mathsf{type} = u)$, each non-transportable term of the form $\{Q^{j_q}[\mathbf{C}'_q] + 1 - \sum_{\mathbf{c}_q} Q^{j_q}[\mathbf{C}'_q]\}$ is not a single probability, which can lead to PATR$(\mathsf{type} = u)$ being greater than one.

**Proposition 8.** *Let $\ell = $ PATR$(y, \mathbf{x}, \mathcal{G}, \boldsymbol{\Delta}, \ell)$ and $u = \min\{1, $PATR$(y, \mathbf{x}, \mathcal{G}, \boldsymbol{\Delta}, u)\}$. Then $P^*_{\mathbf{x}}(y)$. $[\ell, u]$ is partially-transportable with respect to $\langle \mathcal{G}^{\boldsymbol{\Delta}}, \mathbb{Z} \rangle$ with the bounds given by $[\ell, u]$.*

*Proof.* It has been established in Prop. 7 that the expressions returned by PATR$(\mathsf{type} = \ell)$ and PATR$(\mathsf{type} = u)$ are sound. Since $0 < \ell \leq P^*_{\mathbf{x}}(y)$, the lower bound $\ell$ lies within the unit interval. Moreover, since the use of the $\min$ operator ensures that $P^*_{\mathbf{x}}(y) \leq u \leq 1$, the upper bound $u$ lies within $[0, 1]$. Thus, the interval $[\ell, u]$ is strictly contained in $[0, 1]$, constituting a valid partially-transportable bound. $\square$

**Runtime analysis.** The algorithm PATR runs in $\mathcal{O}(zn^4)$ where $z$ is the number of experiments $|\mathbb{Z}|$ and $n$ denote the number of vertices $|\mathbf{V}|$. Lines 3–5 are satisfied only once until the given query is fully factorized. Each factorized sub-query subsequently encounters Lines 3 and 4 once before

proceeding to Line 6. The procedure PAID can be invoked at most $z$ times. Each call to PAID runs in $\mathcal{O}(n^3)$ time. The procedure PATR may be called up to $\mathcal{O}(n)$ times due to the factorization step in Line 6. Each invocation of PATR may call PAID up to $z$ times, resulting in a total of $\mathcal{O}(zn)$ PAID calls. Since each of these calls may trigger recursion up to $n$ times, the total number of recursive steps remains bounded. Assuming that set and graphical operations take $\mathcal{O}(n^2)$ time, the overall runtime is $\mathcal{O}(zn^4)$.

### E.2 CAUSAL BOUNDS OF EXPECTED REWARDS

In this section, we present the algorithm associated with Thm. 2, which maximizes and minimizes over the sources to compute the tightest possible causal bounds, as demonstrated in Alg. 5.

---

**Algorithm 5:** Causal bounds algorithm (CAUSALBOUND).

1 **function** CAUSALBOUND$(y, \mathbf{x}, \mathcal{G}, \boldsymbol{\Delta}, \mathbb{Z}, \mathcal{P}_{\mathbb{Z}}^{\Pi})$

**Input:** $y$: reward; $\mathbf{x}$: action; $\mathcal{G}$: causal diagram; $\boldsymbol{\Delta}$: discrepancies; $\mathbb{Z}$ experiments; $\mathcal{P}_{\mathbb{Z}}^{\Pi}$: distributions.

2      Initialize causal bounds $[\ell_{\mathbf{x}}, u_{\mathbf{x}}]$ as $[0, \infty)$.

3      **for** $P_{\mathbf{x}}^{\ell}(y) := \text{PATR}(\{y\}, \mathbf{x}, \mathcal{G}, \boldsymbol{\Delta}, \ell)$ **do** $\ell_{\mathbf{x}} = \max\{\ell_{\mathbf{x}}, \sum_y y \cdot P_{\mathbf{x}}^{\ell}(y)\}$.

4      **for** $P_{\mathbf{x}}^{u}(y) := \text{PATR}(\{y\}, \mathbf{x}, \mathcal{G}, \boldsymbol{\Delta}, u)$ **do** $u_{\mathbf{x}} = \min\{u_{\mathbf{x}}, \sum_y y \cdot \min\{1, P_{\mathbf{x}}^{u}(y)\}\}$.

5      **return** $[\ell_{\mathbf{x}}, u_{\mathbf{x}}]$.

---

In Lines 3 and 4, the algorithm derives valid expressions for bounds of $P_{\mathbf{x}}^*(y)$ from PATR$(\text{type} = \ell)$ and PATR$(\text{type} = u)$, and updates the final bounds using $\mathcal{P}_{\mathbb{Z}}^{\Pi}$ to obtain tighter estimates. Specifically, if the estimated upper bound exceeds one, we replace it with one to maintain valid probabilistic bounds.

## F    REGRET UPPER BOUNDS OF TRUCB

In this section, we provide detailed proofs of Prop. 2.

**Lemma 6** (Hoeffding's inequality (Hoeffding, 1994)). *Suppose $Y_1, \cdots, Y_n$ are independent random variables such that $Y_i \in [0, 1]$ with $a_i < b_i$ for all $i$. Then, the following holds:*

$$P(|\bar{Y}_n - \mu| \geq \epsilon) \leq 2e^{-2n\epsilon^2}$$

*Therefore, setting $\delta = 2e^{-2n\epsilon^2}$, we get $\bar{Y}_n \leq \mu + \sqrt{\frac{\ln(1/\delta)}{2n}}$ with probability $1 - \delta$.*

**Proposition 2** (Regret bound). *Let $Y$ be the reward variable supported on $[0, 1]$. Then, the cumulative regret of TRUCB in the target SCM $\mathcal{M}^*$ after $T > 1$ rounds is bounded as*

$$R_T \leq 8 \sum_{\mathbf{x}: \Delta_{\mathbf{x}} > 0, \, u_{\mathbf{x}} \geq \mathbb{E}_{P_{\mathbf{x}^*}^*} Y} \frac{\log(T)}{\Delta_{\mathbf{x}}} + \left(1 + \frac{\pi^2}{3}\right) \sum_{\mathbf{x}: \Delta_{\mathbf{x}} > 0, \, u_{\mathbf{x}} \geq \ell^\star} \Delta_{\mathbf{x}}. \tag{9}$$

*Proof.* The proof follows the argument of Auer et al. (2002) and Zhang and Bareinboim (2017). For convenience, we denote the optimal expected reward in the target environment $\mathbb{E}_{P_{\mathbf{x}^*}^*} Y$ by $\mu^\star$. Let $\ell^\star \leq u_{\mathbf{x}} < \mu^\star$. Then, we have

$$N_T(\mathbf{x}) = \sum_{t=1}^{T} \mathbb{I}\{\mathbf{X}_t = \mathbf{x}\} \tag{24}$$

$$\leq \sum_{t=1}^{T} \mathbb{I}\{\bar{U}_{\mathbf{x}^\star}(t) < \mu^\star\} + \mathbb{I}\{\bar{U}_{\mathbf{x}^\star}(t) \geq \mu^\star, \mathbf{X}_t = \mathbf{x}\} \tag{25}$$

$$= \sum_{t=1}^{T} \mathbb{I}\{\bar{U}_{\mathbf{x}^\star}(t) < \mu^\star\} + \mathbb{I}\{\bar{U}_{\mathbf{x}}(t) \geq \mu^\star\}. \tag{26}$$

The last step holds since an arm $\mathbf{x}$ is played at time step $t$ if and only if the causal clipped UCB $\bar{U}_\mathbf{x}(t)$ is maximal. Since $\ell_\mathbf{x} \leq \bar{U}_\mathbf{x}(t) \leq u_\mathbf{x}$ and $u_\mathbf{x} < \mu^\star$, we have $\bar{U}_\mathbf{x}(t) < \mu^\star$. Thus,

$$N_T(\mathbf{x}) \leq \sum_{t=1}^{T} \mathbb{I}\{\bar{U}_{\mathbf{x}^\star}(t) < \mu^\star\} \tag{27}$$

$$\leq \sum_{t=1}^{T} \mathbb{I}\{U_{\mathbf{x}^\star}(t) < \mu^\star\}. \tag{28}$$

Hence, we can derive the following:

$$\mathbb{E}N_T(\mathbf{x}) \leq \sum_{t=1}^{T} P(U_{\mathbf{x}^\star}(t) < \mu^\star) \tag{29}$$

$$\leq \sum_{t=1}^{T} P(\hat{\mathbb{E}}_{P_{\mathbf{x}^\star}^*, t} Y + \sqrt{\frac{2\ln(t)}{N_t(\mathbf{x}^\star)}} < \mu^\star) \tag{30}$$

$$\leq \sum_{t=1}^{T} \sum_{N_t(\mathbf{x}^\star)=1}^{t} P(\hat{\mathbb{E}}_{P_{\mathbf{x}^\star}^*, t} Y + \sqrt{\frac{2\ln(t)}{N_t(\mathbf{x}^\star)}} < \mu^\star) \tag{31}$$

$$\leq \sum_{t=1}^{T} \sum_{N_t(\mathbf{x}^\star)=1}^{t} \delta = \sum_{t=1}^{T} \frac{1}{t^3} \leq \frac{\pi^2}{6} \tag{32}$$

where Eq. (31) to Eq. (32) follows Hoeffding's inequality. Next, **let** us consider $u_\mathbf{x} > \mu^\star$, i.e., the prior upper causal bound is greater than the optimal expected reward. Let $l$ be an arbitrary positive integer.

$$N_T(\mathbf{x}) = \sum_{t=1}^{T} \mathbb{I}\{\mathbf{X}_t = \mathbf{x}\} \tag{33}$$

$$\leq l + \sum_{t=1}^{T} \mathbb{I}\{\bar{U}_{\mathbf{x}^\star}(t) \leq \bar{U}_\mathbf{x}(t), N_t(\mathbf{x}) \geq l\} \tag{34}$$

$$\leq l + \sum_{t=1}^{T} \mathbb{I}\{U_{\mathbf{x}^\star}(t) \leq U_\mathbf{x}(t), N_t(\mathbf{x}) \geq l\} \tag{35}$$

$$= l + \sum_{t=1}^{T} \mathbb{I}\{\hat{\mathbb{E}}_{P_{\mathbf{x}^\star}^*, t} Y + \sqrt{\frac{2\ln(t)}{N_t(\mathbf{x}^\star)}} \leq \hat{\mathbb{E}}_{P_\mathbf{x}^*, t} Y + \sqrt{\frac{2\ln(t)}{N_t(\mathbf{x})}}, N_t(\mathbf{x}) \geq l\} \tag{36}$$

$$\leq l + \sum_{t=1}^{T} \sum_{N_t(\mathbf{x}^\star)=1}^{t} \sum_{N_t(\mathbf{x})=l}^{t} \mathbb{I}\{\hat{\mathbb{E}}_{P_{\mathbf{x}^\star}^*, t} Y + \sqrt{\frac{2\ln(t)}{N_t(\mathbf{x}^\star)}} \leq \hat{\mathbb{E}}_{P_\mathbf{x}^*, t} Y + \sqrt{\frac{2\ln(t)}{N_t(\mathbf{x})}}\} \tag{37}$$

The last event implies that at least one of the following events occur:

$$\hat{\mathbb{E}}_{P_{\mathbf{x}^\star}^*, t} Y + \sqrt{\frac{2\ln(t)}{N_t(\mathbf{x}^\star)}} \leq \mathbb{E}_{P_{\mathbf{x}^\star}^*} Y \tag{38}$$

$$\hat{\mathbb{E}}_{P_\mathbf{x}^*, t} Y - \sqrt{\frac{2\ln(t)}{N_t(\mathbf{x})}} \geq \mathbb{E}_{P_\mathbf{x}^*} Y \tag{39}$$

$$\mathbb{E}_{P_{\mathbf{x}^\star}^*} Y < \mathbb{E}_{P_\mathbf{x}^*} Y + 2\sqrt{\frac{2\ln(t)}{N_t(\mathbf{x})}} \tag{40}$$

Remaining part follows the proof of Theorem 1 in Auer et al. (2002). We bound the probability of events Eqs. (38) and (39) using Hoeffding's inequality (Hoeffding, 1994). Eq. (40) does not appear

$$\mathbb{E}_{P_{\mathbf{x}^\star}^*} Y - \mathbb{E}_{P_\mathbf{x}^*} Y - 2\sqrt{\frac{2\ln(t)}{N_t(\mathbf{x})}} = \Delta_\mathbf{x} - 2\sqrt{\frac{2\ln(t)}{N_t(\mathbf{x})}} \geq 0 \tag{41}$$

for $N_t(\mathbf{x}) \geq \frac{8\ln t}{(\Delta_\mathbf{x})^2}$. Therefore, with $l \geq \left\lceil \frac{8\ln t}{(\Delta_\mathbf{x})^2} \right\rceil$, we get

$$\mathbb{E}N_T(\mathbf{x}) \leq \left\lceil \frac{8\ln t}{(\Delta_\mathbf{x})^2} \right\rceil + \sum_{t=1}^{T} \sum_{N_t(\mathbf{x}^\star)=1}^{t} \sum_{N_t(\mathbf{x})=\lceil 8\ln t/(\Delta_\mathbf{x})^2 \rceil}^{t} 2t^{-4} \leq \frac{8\ln t}{(\Delta_\mathbf{x})^2} + 1 + \frac{\pi^2}{3} \tag{42}$$

| Total trials | Task 1
50k | Task 2
50k | Task 3
50k |
|---|---|---|---|
| TRUCB | **47.94** $\pm$ 9.60 (36.83%) | **85.25** $\pm$ 113.3 (38.62%) | **95.69** $\pm$ 7.47 (55.11%) |
| POUCB | $130.16 \pm 7.33$ | $220.75 \pm 117.82$ | $173.62 \pm 7.56$ |
| UCB | $397.03 \pm 12.33$ | $220.75 \pm 117.82$ (= CR of POUCB) | $214.42 \pm 7.76$ |

Table 1: Mean and standard deviation of cumulative regret over 1,000 repeated simulations. The percentages (red) represent the ratio $\frac{\text{CR for TRUCB}}{\text{CR for POUCB}} \times 100(\%)$.

which concludes the proof. $\qquad\square$

## G DETAILS ON EXPERIMENTAL SETTINGS

We compare TRUCB (Alg. 1) against standard UCB over all combinations of arms (UCB), defined as $\{\mathbf{x} \in \Omega_{\mathbf{X}} \mid \mathbf{X} \in 2^{\mathbf{V}\setminus\{Y\}\setminus\mathbf{N}^*}\}$ and over POMISs (POUCB), defined as $\{\mathbf{x} \in \Omega_{\mathbf{X}} \mid \mathbf{X} \in \mathbb{P}_{\mathcal{G},Y}^{\mathbf{N}^*}\}$. Our baseline comparison is with POUCB, to ensure a fair evaluation of transportability performance. The number of trials is set to 50,000, which is sufficient to observe the performance differences. Each simulation is repeated 1,000 times to ensure consistent results. The experiments were conducted on a Linux server equipped with an Intel Xeon Gold 5317 processor running at 3.0 GHz and 64 GB of RAM. No GPUs were used during the simulations.

### G.1 DETAILED EXPLANATIONS OF THE WORKING EXAMPLES

We provides details on the workings of the TRUCB algorithm (Alg. 1) and the specific SCMs used in all bandit instances presented in the experiments (Sec. 5). We denote the exclusive-or operation by $\oplus$, and use Bern to represent a Bernoulli distribution. We *randomly* generate structural functions $\mathbf{F}$ using binary logical operations ($\wedge, \vee, \oplus, \neg$), and the parameters of the exogenous variable distributions are also *randomly* selected. The following table (Table 1) summarizes our simulation results.

**Task 1.** The bandit instance is associated with an SCM $\mathcal{M}$ where

$$
\mathcal{M} = \begin{cases}
\mathbf{U} & = \{U_A, U_B, U_Y, U_{AY}\} \\
\mathbf{V} & = \{A, B, C, Y\} \\
\mathbf{F} & = \begin{cases} f_A = u_A \vee (b \oplus u_{AY}), f_B = u_B, \\ f_Y = (1-b) \vee ((u_{AY} \vee b) \oplus a)\} \end{cases} \\
P(\mathbf{U}) & = \begin{cases} U_A \sim \text{Bern}(0.33), U_B \sim \text{Bern}(0.38), \\ U_Y \sim \text{Bern}(0.41), U_{AY} \sim \text{Bern}(0.75). \end{cases}
\end{cases}
\tag{43}
$$

The decision maker has an experimental prior $\mathbb{Z}^1 = \{\{B\}\}$ from $\Delta^1 = \{A\}$. The action space without prior information corresponds to $\mathbb{P}_{\mathcal{G},Y}^{\mathbf{N}^*=\emptyset} = \{\{B\}, \{A, B\}\}$, which corresponds to the action space of the baseline algorithm POUCB (i.e., the initialized target action space). In this setting, we observe that $\mathbb{E}_{P_b^*} Y$ cannot be bounded from the source $\pi^1$, since $\{A\} \cap \Delta^1 \neq \emptyset$. In contrast, $\mathbb{E}_{P_{a,b}^*} Y$ can be bounded as $\sum_y y P_b^1(a, y) \leq \mathbb{E}_{P_{a,b}^*} Y \leq \sum_y y\{P_b^1(a, y) + 1 - P_b^1(a)\}$. Using these expressions, the decision maker can estimate causal bounds for four actions corresponding to the POMISs: $do(A = 0, B = 0) : [0.1675, 1]$, $do(A = 0, B = 1) : [0.2965, 0.7940]$, $do(A = 1, B = 0) : [0.8325, 1]$ and $do(A = 1, B = 1) : [0.2935, 0.7960]$. The algorithm computes dominance bounds as $\ell^* = 0.8325$ and $u^* = \infty$. Here, $u^* = \infty$ arises because the causal bounds for $do(b)$ are $[0, \infty)$, i.e., no information can be transported for that intervention. Among the four actions, the upper causal bounds for $do(A = 0, B = 1)$ and $do(A = 1, B = 1)$ are lower than $\ell^*$, leading them to be excluded by TRUCB. Accordingly, the algorithm begins the online interaction with the final action space $\mathcal{I}^*$ excluding these two actions. We observe that the mean cumulative regret at the final trial is **47.94** ($\pm$ 9.60) for TRUCB and **130.16** ($\pm$ 7.33) for POUCB, which is **36.83**% of the latter.

**Task 2.** The bandit instance is associated an SCM $\mathcal{M}$ where

$$\mathcal{M} = \begin{cases} \mathbf{U} & = \{U_A, U_B, U_C, U_Y, U_{BC}, U_{BY}\} \\ \mathbf{V} & = \{A, B, C, Y\} \\ \mathbf{F} & = \begin{cases} f_A = u_A, f_B = (u_{BY}) \wedge u_{BC}) \oplus u_B), \\ f_C = b \oplus (u_{BC} \wedge ((1 - u_C) \oplus a)), \\ f_Y = c \oplus (u_{BY} \wedge ((1 - u_Y) \oplus a)) \end{cases} \\ P(\mathbf{U}) & = \begin{cases} U_A \sim \texttt{Bern}(0.28), U_B \sim \texttt{Bern}(0.42), U_C \sim \texttt{Bern}(0.52) \\ U_Y \sim \texttt{Bern}(0.47), U_{BC} \sim \texttt{Bern}(0.62), U_{BY} \sim \texttt{Bern}(0.49). \end{cases} \end{cases} \tag{44}$$

An agent has priors from two environments with potential discrepancies $\Delta^1 = \{A\}$ and $\Delta^2 = \{B\}$. The prior from the first source is observational $\mathbb{Z}^1 = \{\emptyset\}$, and from the second source is experimental $\mathbb{Z}^2 = \{\{C\}\}$. The non-manipulable variable constraint is $\mathbf{N}^* = \{A, C\}$, and the corresponding initial action space is $\mathbb{P}_{\mathcal{G}, Y}^{\mathbf{N}^*} = \{\emptyset, \{B\}\}$. The action $do(\emptyset)$ is transportable as $\mathbb{E}_{P_\emptyset^*} Y = \sum_{a,b,c,y} y Q^2[A] Q^1[B, C, Y] = \sum_{a,b,c,y} y P_c^2(a) P^1(b, c, y \mid a) = 0.4844$. On the other hand, $do(b)$ is *not* transportable, since $Q^*[C]$ is *non-transportable* from the sources: $P_b^*(y) = \sum_{a,c} Q^2[A] Q^2[Y] Q^*[C]$. Since $\Delta^1 \cap \{C\} = \emptyset$ and according to Thm. 2, $Q^*[C]$ is bounded by $Q^1[B, C]$, which leads to $\ell_b = \sum_{a,c,y} y P_c^2(a) P_c^2(y|a) P^1(c|a, b) P^1(b)$. This expression yields $\ell_{do(B=0)} = 0.2097$ and $\ell_{do(B=1)} = 0.2752$. The upper causal bound for $do(b)$ is given by $\sum_y y \min\{1, \sum_{a,c} P_c^2(a) P_c^2(y|a)\{P^1(b) P^1(c|a, b) + 1 - P^1(b)\}\}$. We thus have $u_{do(B=0)} = 0.6783$ and $u_{do(B=1)} = 0.8066$. We now proceed to compute the dominance bounds. The set of MISs with respect to $\langle \mathcal{G}, Y, \mathbf{N}^* \rangle$ is $\{\emptyset, \{B\}\}$. Since $\ell_\emptyset = 0.4844$ and $\ell_{b^*} = 0.6783$, we have the dominance lower bound is given by $\ell^* = 0.4844$. To compute the upper dominance bound $u^*$, the algorithm (executed via the subroutine UDB in Alg. 3) initializes with 0.8066 (i.e., the upper causal bound $u_{do(B=1)}$) and traverses weaker constraints (see Fig. 9): (b) For $\mathbf{N} = \{A\}$, the bound remains 0.8066. (c) For $\mathbf{N} = \{C\}$, the algorithm returns 0.8070, which corresponds to the upper causal bound $u_{do(A=0, B=1)}$. (d) For $\mathbf{N} = \emptyset$, the algorithm returns 0.7697, which corresponds to the expected reward of the transportable action $do(A = 1, C = 1)$—$\mathbb{E}_{P_{do(A=1, C=1)}^*} Y = \sum_y y P_{do(C=1)}^2(y|A=1) = 0.7697$.

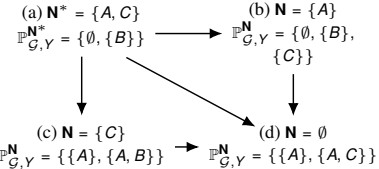

(a) $\mathbf{N}^* = \{A, C\}$
$\mathbb{P}_{\mathcal{G}, Y}^{\mathbf{N}^*} = \{\emptyset, \{B\}\}$

(b) $\mathbf{N} = \{A\}$
$\mathbb{P}_{\mathcal{G}, Y}^{\mathbf{N}} = \{\emptyset, \{B\}, \{C\}\}$

(c) $\mathbf{N} = \{C\}$
$\mathbb{P}_{\mathcal{G}, Y}^{\mathbf{N}} = \{\{A\}, \{A, B\}\}$

(d) $\mathbf{N} = \emptyset$
$\mathbb{P}_{\mathcal{G}, Y}^{\mathbf{N}} = \{\{A\}, \{A, C\}\}$

Figure 9: Hierarchical relationships between POMISs under different constraints. Arrows indicate the direction of dominance relations.

Consequently, the final upper dominance bound is set to $u^* = 0.7697$. Since $u_{do(B=1)} = 0.8066 > 0.7697 = u^*$, the final transport upper bound is updated to the value of $u^* = 0.7697$. The resulting transport bounds for $do(B=0)$ and $do(B=1)$ are $[0.2097, 0.6783]$ and $[0.2752, 0.7697]$, respectively. Although the size of the action space remains unchanged (i.e., no action is removed from $\mathcal{I}^*$, implying that the action spaces of all three algorithms—UCB, POUCB and TRUCB—are identical), we observe that accounting for the transport bounds improves performance, with final mean regrets of **220.75** ($\pm 117.82$) for POUCB and **85.25** ($\pm 113.3$) for TRUCB, corresponding to **38.62%** of the baseline algorithm POUCB.

**Task 3.** The bandit instance is associated an SCM $\mathcal{M}$ where

$$\mathcal{M} = \begin{cases} \mathbf{U} & = \{U_R, U_T, U_W, U_X, U_Z, U_Y, U_{RW}, U_{RY}, U_{XY}, U_{WX}, U_{WZ}\} \\ \mathbf{V} & = \{R, T, W, X, Z, Y\} \\ \mathbf{F} & = \begin{cases} f_R = u_R \vee ((1 - u_{RW}) \wedge (t \wedge u_{RY})), f_T = u_T \\ f_W = u_W \vee ((1 - u_{WX}) \wedge (u_{RW} \wedge u_{WZ})), \\ f_X = w \vee ((1 - u_{WX}) \wedge (u_{XY} \wedge u_X)), \\ f_Z = z \vee ((1 - u_{XY}) \wedge (u_{RY} \wedge u_Y)), \\ f_Y = x \vee ((1 - u_{WZ}) \wedge (r \wedge u_Z)) \end{cases} \\ P(\mathbf{U}) & = \begin{cases} U_R \sim \texttt{Bern}(0.53), U_T \sim \texttt{Bern}(0.63), U_W \sim \texttt{Bern}(0.38), \\ U_X \sim \texttt{Bern}(0.27), U_Z \sim \texttt{Bern}(0.4), U_Y \sim \texttt{Bern}(0.26), \\ U_{RW} \sim \texttt{Bern}(0.52), U_{RY} \sim \texttt{Bern}(0.63), U_{XY} \sim \texttt{Bern}(0.79), \\ U_{WX} \sim \texttt{Bern}(0.74), U_{WZ} \sim \texttt{Bern}(0.31). \end{cases} \end{cases} \tag{45}$$

An agent has access to priors from two environments with potential discrepancies $\Delta^1 = \{R\}$ and $\Delta^2 = \{T\}$, with priors $\mathbb{Z}^1 = \{\emptyset, \{Z\}\}$ and $\mathbb{Z}^2 = \{\{Z\}\}$, and constraint $\mathbf{N}^* = \{T, W\}$. The initial action space is $\mathbb{P}_{\mathcal{G}, Y}^{\mathbf{N}^*} = \{\emptyset, \{R\}, \{X\}, \{Z\}\}$ where $\mathbb{E}_{P_\emptyset^*} Y = \sum_{r,t,w,x,z,y} y P^1(r, x, y, z | t, w) P^1(w) P_z^2(t)$ and $\mathbb{E}_{P_z^*} Y = \sum_y y P_z^1(y) = \sum_y y P_z^2(y)$ are transportable while $\mathbb{E}_{P_x^*} Y = \sum_{r,t,z,y} y Q^1[R, Y] Q^2[T] Q^*[Z]$ and $\mathbb{E}_{P_r^*} Y = \sum_{w,x,z,y} y Q^*[W, X, Z, Y]$ are *not*. According to Thm. 2, we have that $Q^*[Z]$ can be bounded by $Q^1[W, R, X, Z, Y] = P^1(w, r, x, z, y \mid t)$, leading to $\sum_{r,t,z,y} y P_z^1(r, y | t) P_z^2(t) P^1(r, w, x, z | t) \leq \mathbb{E}_{P_x^*} Y \leq \sum_y y \min\{1, \sum_{r,t,z} P_z^1(r, y | t) P_z^2(t)(P^1(r, w, x, z | t) + 1 - P^1(r, w, x | t))\}$. Similarly, we can derive the causal bounds for $do(r)$ using $Q^1[W, R, X, Z, Y]$ as $\sum_{w,x,z} P^1(r, x, y, z | t, w) P^1(w) \leq \mathbb{E}_{P_r^*} Y \leq \sum_y y \min\{1, \sum_{w,x,z} P^1(r, x, y, z | t, w) P^1(w) + 1 - P^1(r, x, z | t, w) P^1(w))\}$. In this setting, the transportable target POMIS action yields $\mathbb{E}_{P_{do(Z=1)}^*} Y = 1$, resulting in $\ell^\star = u^\star = 1$. Furthermore, we find the upper causal bounds $u_{do(\emptyset)} = 0.5514$, $u_{do(X=0)} = 0.7901$ and $u_{do(Z=0)} = 0.034$—these three upper causal bounds are lower than $\ell^\star$. Hence, the corresponding actions are eliminated from $\mathcal{I}^*$ by the algorithm. We observe mean cumulative regrets of **173.62** ($\pm 7.56$) for POUCB and **95.69** ($\pm 7.47$) for TRUCB, corresponding to **55.11%** of regret ratio.

## H DISCUSSIONS

**Misspecification.** In the transportability literature, assuming access to true selection diagram is a standard modeling practice. As noted by Bareinboim and Pearl (2016), *if no knowledge about commonalities and disparities across environments is available, transportability cannot be justified*. That said, some degree of misspecification can be tolerated without invalidating the performance guarantees—particularly when the assumed causal diagram or selection diagram forms *super-model* of the true environment. A similar discussion was presented by Bellot et al. (2023).

For example, suppose both the source and target environments share the causal diagram $\mathcal{G} = \langle \{X, Y\}, \{X \to Y\} \rangle$. Given source data $P(x, y)$ and the graph structure, the interventional distribution $P_x(y) = P(y \mid x)$ is identifiable. However, if we are unsure about the presence of an unobserved confounder, we can conservatively posit a super-model $\mathcal{G}' = \langle \{X, Y\}, \{X \to Y, X \leftrightarrow Y\} \rangle$. Under this model, while point identification fails, the interventional distribution $P_x(y)$ is bounded within the interval $[P(x, y), P(x, y) + 1 - P(x)]$, which still contains the true value. The same reasoning applies to selection diagrams. The selection nodes only indicate potential discrepancies between environments. If a researcher is uncertain about whether a mechanism differs across environments, they may still conduct valid inference by conservatively assuming the presence of a discrepancy. Such conservatism does not harm transportability guarantees. Importantly, this conservative modeling increases the number of POMISs. For instance, a POMIS under $\mathcal{G}$ is $\{\{X\}\}$, while under $\mathcal{G}'$ it becomes $\{\emptyset, \{X\}\}$, which covers the true POMIS. This leads to less informative but still correct inferences, outperforming methods that ignore structural information altogether.

In contrast, misspecifying in the opposite direction (i.e., failing to model a discrepancy that does exist in the target environment) can lead to incorrect inferences. This reflects a fundamental asymmetry: being conservative preserves soundness, but missing edges or selection nodes can violate correctness.

**Parametric approach.** One might be concerned that while the algorithm (Alg. 1) effectively uses prior knowledge to eliminate non-optimal actions before learning begins, it then switches to a traditional UCB approach that ignores additional observations available during each round. Indeed, there exists a rich body of research that incorporates prior knowledge to iteratively update parameters of SCMs under graphical constraints (Zhang and Bareinboim, 2022; Bellot et al., 2023; Wei et al., 2023; Jalaldoust et al., 2024) in online learning. However, such approaches often rely on optimization-based approaches—such as *canonical SCM* (Zhang et al., 2022) or *neural causal models* (NCMs) (Xia et al., 2021) that assume full parameterization, which results in high computational overhead. This approach is infeasible for larger or denser causal diagrams, and exploiting them effectively remains an open problem (Elahi et al., 2024). Moreover, these methods are constrained to categorical settings, which makes them inapplicable to continuous domains, such as those encountered in causal

Bayesian optimization (CBO; Aglietti et al. (2020) and muti-outcome variant MO-CBO Bhatija et al. (2025)), where POMISs are also leveraged for structural pruning in continuous action spaces.

In contrast, our approach focuses on leveraging structural knowledge offline, before any online interaction, and without requiring parameterization or any strong assumptions beyond a given graphical structure and sources. This distinction allows us to scale to settings where parameter learning is infeasible or computationally prohibitive (dense graph or continuous domain), while still retaining provable regret guarantees through structure-informed pruning and closed-form bounding.

## I  LIMITATIONS

In this section, we discuss limitations of our work and outline promising directions for future research.

**Modeling bandit instances in the form of SCMs.**   Structural Causal Models (SCMs) are a versatile and expressive framework that provides a principled way to represent and reason about causal relationships. Their generality makes them applicable across a wide range of domains. However, SCMs come with certain limitations, such as the assumption of a well-defined set of variables and a fixed causal structure, which may not adequately capture the complexity of dynamic, high-dimensional, or partially observed systems. Nonetheless, our work addresses a fundamental problem within the SCM framework. We believe it provides a solid foundation for future research, such as extending causal bandits to more complex or less structured environments.

**Known causal diagram.**   We make the standard assumption that the deployment learner has access to the underlying causal diagram. While knowledge of the causal structure can greatly enhance decision-making, this requirement may limit the broader applicability of the proposed approach. In practice, several techniques—such as causal discovery methods or the use of ancestral graphs as plausible explanations—can help alleviate this issue. However, these techniques typically require substantial domain knowledge, and thus, the assumption remains a key limitation of our framework.

While a collective selection diagram provides a principled and interpretable framework for analyzing environment shifts and transferring information across environments, it arguably restricts the analysis to a narrow class of problems where common (super) causal diagrams should be explicitly defined. Consequently, the applicability of this framework may be limited in settings where environmental changes substantially alter the underlying graph structure.

**Tightness of causal bounds.**   As discussed in Appendix E, the upper bound computed by the algorithm PATR (Alg. 4) can, in some cases, exceed one. Moreover, we do not guarantee the *tightness* of our causal bounds; that is, we cannot ensure the existence of an SCM under which the causal effect exactly equals either $\ell$ or $u$. Characterizing conditions under which lower and upper bounds are attainable by some SCM would enhance the interpretability and reliability of our framework. Investigating tighter bounds and formally establishing their tightness remains an important direction for future work.

**Sufficient prior data from source environments.**   Our method relies on the availability of prior data to construct dominance and causal bounds for guiding exploration. However, when the prior data is insufficient or biased, the resulting bounds may become inaccurate. In particular, inaccurate bounds may fail to include the true expected reward of certain actions. As a result, the agent may prematurely eliminate potentially optimal actions from exploration, leading to suboptimal performance. Moreover, our current framework focuses on the transportability of causal quantities, rather than on improving estimation quality under noisy or limited prior data. Developing robust algorithms that can explicitly account for uncertainty in the prior and avoid overconfident pruning remains an important direction for future work.

IMPACT STATEMENT

This work addresses a structured causal bandit framework that leverages prior knowledge from heterogeneous environments through transportability theory in the causal inference literature. This approach has potential applications in real-world settings where experimentation is costly or limited, such as personalized healthcare, adaptive education, and resource-constrained recommendation systems. In these domains, improper specification of causal structures may lead to misleading conclusions and biased decisions. Therefore, careful validation and domain-specific causal modeling are essential before deployment in high-stakes environments.

