# OpenReview forum: "On Transportability for Structural Causal Bandits"
_ICLR.cc/2026/Conference — ICLR 2026 Conference Withdrawn Submission_

### Official Review · Reviewer_Zeoe · 2025-10-27

**Soundness:** 4
**Presentation:** 3
**Contribution:** 3
**Rating:** 6
**Confidence:** 3

**Summary:**

The paper introduces the structural causal bandit framework with transportability, enabling the fusion of priors from heterogeneous source environments to enhance learning in a target environment. It establishes hierarchical relations among action spaces and derives dominance bounds on expected rewards. It proposes a method to estimate expected rewards or their causal bounds using arbitrary observational or experimental data from sources. It presents a UCB-based algorithm (TRUCB) that incorporates causal knowledge to guide exploration, achieving a sub-linear cumulative regret bound depending on the informativeness of prior data, with empirical results showing improved performance over baselines.

**Strengths:**

1. Hierarchical Dominance Relations in Action Spaces: The paper derives dominance bounds for expected rewards, such as $E P^_{x^\star} Y \leq E P^_{r^\star} Y$ where $r^\star$ is optimal under weaker constraints, allowing efficient pruning of non-optimal actions like non-POIS sets without full exploration.
2. Transportability for Causal Bounds: It introduces a method using c-factors $Q^*[C_q]$ to bound non-transportable expected rewards, e.g., $\ell = Q^i[C']$ and $u = Q^i[C'] + 1 - \sum_c Q^i[C']$ for c-components $C \subseteq C'$, enabling partial knowledge transfer from heterogeneous sources.
3. UCB Algorithm with Causal Priors: The TRUCB algorithm clips UCB indices using offline-computed dominance bounds, leading to a regret bound $R_T = \sum_x \Delta_x \mathbf{E} N_T(x)$ that explicitly depends on prior informativeness, reducing online exploration costs.
4. Graphical Encoding of Discrepancies: The collective selection diagram $G_\Delta$ with selection nodes $S$ models environment shifts, allowing fusion of observational and interventional data via do-calculus rules to compute transportable quantities like $P^*a(y) = \sum{b,c} P^1_\emptyset(c | a, b) P^2_{a,c}(b, y)$.

**Weaknesses:**

1. Lack of Regret Lower Bound: The paper provides an upper bound on cumulative regret but does not derive a corresponding lower bound, limiting assessment of the algorithm's theoretical optimality in the structural causal bandit setting.
2. Straightforward Combination of Frameworks: The core model integrates structural causal models (SCMs) with transportability via selection diagrams in a direct manner, where dominance relations largely extend POMIS definitions without introducing fundamentally new graphical criteria.
3. Simplistic Online Algorithm Phase: The TRUCB relies on clipping UCB values based on precomputed bounds from offline transportability analysis, resembling a basic nesting of UCB with causal pruning rather than incorporating dynamic online adaptations to observations.

**Questions:**

1. Why is there no regret lower bound provided in the paper? What do you think are the main technical difficulties in deriving such a lower bound, and do you consider constructing the lower bound harder than the upper bound? Do you plan to derive one to assess the tightness of the upper bound?
2. How could the framework be extended to handle continuous variables or partially unknown causal graphs, as mentioned in the limitations? What are the key challenges in such extensions, and are there potential approaches you envision for addressing them?

---

### Official Review · Reviewer_1314 · 2025-10-31

**Soundness:** 3
**Presentation:** 3
**Contribution:** 2
**Rating:** 4
**Confidence:** 3

**Summary:**

This paper extends the traditional structural causal bandit framework to settings where data from multiple environments (SCMs) can be leveraged to improve online learning in a target domain. Building on Pearl’s theory of transportability, it formalizes how causal knowledge obtained from different source environments can be transferred under specific graphical conditions expressed through selection diagrams. To incorporate this information, the authors propose the TC-UCB algorithm, which modifies the standard UCB approach by integrating transportable priors and causal dominance relations. The method is proven to achieve a sub-linear regret bound that depends explicitly on an informativeness index capturing both the amount and quality of transportable data. Experiments on synthetic datasets demonstrate notable improvements in the algorithm’s learning efficiency.

**Strengths:**

Please find the strengths below:
1. The paper integrates causal transportability theory with the structural causal bandit framework, enabling the use of information observed in other SCMs to help estimate the effects of interventions. This represents a creative and unconventional approach to improving online learning efficiency.
2. The theoretical analysis is rigorous: the paper formally derives the conditions for transferability and establishes a sub-linear regret bound that depends explicitly on an informativeness index, which quantifies the amount and quality of transferable information.

**Weaknesses:**

Please find the weaknesses below:
1. The approach requires full knowledge of the causal and selection diagrams to determine transportability, which is often unrealistic in practice. The proposed TC-UCB mainly extends causal UCB by adding transportable priors, so its algorithmic novelty is limited.
2. Theoretical analysis does not specify worst-case dependence on the graph size or action space. When the number of interventions is large, the regret may scale exponentially, and there is no discussion of cases where transferability fails—potentially leading to worse performance than standard UCB.
3. Experiments are restricted to small synthetic causal environments, without evaluation on real-world or partially unknown graphs, making empirical evidence insufficient to confirm robustness and scalability.

**Questions:**

The questions are related to the weaknesses:
1. The method assumes full knowledge of the causal and selection diagrams. Could the proposed framework be extended to settings where the causal structure is partially unknown, similar to Feng, Xiong, and Chen (ACML 2024) “Combinatorial Causal Bandits without Graph Skeleton,” and Yan and Tajer (NeurIPS 2024) “Linear Causal Bandits: Unknown Graph and Soft Interventions”?
2. The theoretical analysis does not provide a worst-case characterization of the regret bound with respect to graph size or number of interventions. Could the authors derive or discuss such worst-case dependence to clarify scalability as the causal graph grows?
3. Experiments are conducted only on small synthetic environments. Could the authors test the algorithm on larger or real-world causal datasets, such as those in CausalBench: A Benchmark for Causal Structure Learning and Reasoning (arXiv:2502.06577), to better demonstrate robustness and practical relevance?

---

### Official Review · Reviewer_z35f · 2025-10-31

**Soundness:** 3
**Presentation:** 3
**Contribution:** 1
**Rating:** 2
**Confidence:** 4

**Summary:**

The paper applies transfer learning to improve the learning performance in structural causal bandits. In addition to standard structural causal bandits, the model provides probability distributions from other causal domains, described by a selection diagram. The proposed approach is built upon possibly-optimal minimal intervention sets (POMISs) (Lee and Bareinboim, 2018; 2019) and transportability from multiple source domains (Lee et al., 2020). By using multi-domain knowledge, the expected reward for a certain intervention can be bounded, allowing the learning algorithm to reduce the action space further and set constraints on estimands. The learning performance can be improved.

**Strengths:**

- The presentation is good with intuitive examples.
- Experiment sections show the proposed approach surpasses baselines, which have no information from other domains.

**Weaknesses:**

I find the paper somewhat incremental, and the contribution appears to be relatively limited in scope.

- The transfer learning algorithm design idea is the same as Zhang and Bareinboim (2017), where they use the observational distribution to bound the expected reward. The UCB bandit approach is also the same. In Zhang and Bareinboim (2017), the Thompson sampling approach is also applied; however, it is not utilized in this paper.

- To reduce the action space, the paper applies the POMIS approach (Lee and Bareinboim, 2018; 2019), which is a standard result in causal bandit literature.

- The transportability result is from  (Lee et al., 2020).

- The main theoretical contribution lies in the causal bounds presented in Section 4.1; however, as noted in the limitations, the tightness of these bounds is not guaranteed.

- With causal bounds, some arms can be eliminated from the POMISs. With this reduced action space, the regret upper bound for the proposed algorithm is standard in bandit literature.

**Questions:**

Could you please elaborate on the main technical challenges addressed and the theoretical contributions of this work?

---

### Official Review · Reviewer_VvyE · 2025-10-31

**Soundness:** 2
**Presentation:** 2
**Contribution:** 2
**Rating:** 2
**Confidence:** 3

**Summary:**

This paper proposes two new tools that extend existing ideas, specifically the notion of a minimum intervention set and causal bounding using C-factors,  from the causal bandit literature to the transportability setting. A cumulative regret bound as empirical validations are provided.

**Strengths:**

Introduction to the causality elements is complete and well-compressed, and I appreciate the difficulty in making such a causality-rich paper appealing to a general audience. The problem is interesting and applying bandit ideas to the transportability literature has a lot of potential.

**Weaknesses:**

First, it was difficult to know what was the contribution of the paper from the text; a lot is introduced, but the novel ideas are not clear (or at least what is novel and what is not is unclear).  For example, the paper does not make clear that the definition of transportability in bandits is not novel. It has been studied before under the same notion of regret (e.g. in Bellot, et al., 2023).

Second, it is hard to judge the size of the contribution. The paper combines existing ideas on existing problems, and from what I can see, the theorems and regret bounds follow naturally from the assumptions and trivial probabilities. The authors never combine the two main ideas (using bounds and using a hierarchy).

The algorithm avoids the most interesting aspect of the problem; the interaction between actions played and information gain. We could have scenarios where it is better to play certain actions that are suboptimal because they can reveal a lot about the optimal action (such as in the partial monitoring literature).

The presentation is overwhelming and reads like a survey of the causal bandit literature more than text which is specifically written to explain an algorithm. Giving the reader some hierarchy of concepts could help parse all the notation, which will be difficult to understand for anyone without a background in the literature. For example, the notion that your technique for bounding revolves around finding and bounding C-factor should be described early so the reader knows what techniques are important. Instead, it is first referred to on page 6 after the reader has slogged through 5 pages of notation, examples, and definitions.

Some definitions are confusing. For example, in the definition of identifiable, it is not clear that the definition of identifiability of C means that there exists a C’ with certain properties.  Additionally, Ln. 257: what is transferable bound? What is a divergent path?

Should make it clear that you are bounding the expected regret, not the realized regret.

I have a few technical questions about the regret bound as well.

**Questions:**

How does your regret compare to the typical UCB bound where nothing is known about the population?

How does the scale of the problem come into play? What concentration inequality do you used to verify the estimates are correct, and what assumption do they require about the random variable of the problem? Unless I missed something, I didn’t see any assumed boundedness or tail control. Even if the regret bound scales with \Delta_x^{-1}, you would still need some notion of scale to tune the learning rate. Where is this coming from?

---

### Note · Authors · 2025-11-25

**Comment:**

After carefully considering the reviewers’ feedback and discussing within the author team, we agree with the reviewers’ comments and suggestions, and have decided to withdraw the paper in order to substantially revise and extend the work. We would like to clarify our main contributions as follows.

1. **Partial Transportability: Symbolic and Closed-form Bounds.** To compute causal bounds, we develop a valid and efficient algorithm for deriving closed-form bound expressions for causal effect from arbitrary experimental data collected from heterogeneous sources. While existing works [Zhang and Bareinboim, 2022; Bellot et al., 2023; Jalaldoust et al., 2024] have proposed methods to bound causal effects, they primarily rely on optimization-based approaches—such as *canonical SCM* [Zhang et al., 2022] or *Neural Causal Models* (NCMs) [Xia et al., 2021]—-that assume full parameterization and typically require categorical observational variables. In contrast, our proposed procedure (paTR in Algorithm 4) requires no assumptions beyond the given graphical structure and efficiently (i.e., $\mathcal{O}(z\vert \mathbf{V} \vert^4)$ where $z$ is the number of experiments) yields sound bounds for a given query. Subsequently, CausalBound (Alg. 5) identifies the tightest among these bounds, enabling precise yet tractable offline guidance in the structural causal bandit setting.
2. **Dominance Relations.** While [Lee and Bareinboim, 2018] introduced the notion of POMIS, and [Lee and Bareinboim, 2019] extended this framework under the non-manipulable constraint $\mathbf{N}$, our work is the first to construct dominance relation among POMISs under distinct constraints. To establish this, we first prove that “*any constrained POMIS is an unconstrained MIS”* (Lemma 4). Building on this, we generalize to the key theoretical result that “*POMISs are dominated by POMISs under strictly weaker constraints”* (Theorem 1)*.* We further develop an efficient algorithm that hierarchically traverses the space of constrained POMISs and terminates early when a valid dominance bound is found, thus avoiding unnecessary computation.

We acknowledge that our intentions may not have been communicated clearly enough, and we have therefore decided to withdraw the paper to thoroughly rewrite and clarify the presentation. We sincerely appreciate the reviewers’ constructive feedback and the committee’s time and effort, and we plan to resubmit a significantly improved version in the future.

**Withdrawal Confirmation:**

I have read and agree with the venue's withdrawal policy on behalf of myself and my co-authors.